# Research on Integrated Control Strategy for Wind Turbine Blade Life

**DOI:** 10.3390/s24175729

**Published:** 2024-09-03

**Authors:** Bairen An, Jun Liu, Zeqiu Zhang

**Affiliations:** School of Automation, Xi’an University of Technology, Xi’an 710048, China; 1190310008@stu.xaut.edu.cn (B.A.); 2200321215@stu.xaut.edu.cn (Z.Z.)

**Keywords:** wind turbine blades, crack propagation, remaining useful life

## Abstract

Wind turbine blades bear the maximum cyclic load and varying self-weights in turbulent wind environments, which accelerate the propagation of cracks that ultimately progress from minor faults, resulting in blade failure and significant maintenance and shutdown costs. To address this issue, this paper proposes an adaptive control strategy for the blade’s useful life. The control system is divided into the inner control loop and the outer control loop. The outer loop is based on the Paris crack propagation model combined with a particle filtering algorithm and calculates the degradation of the blade life under the crack threshold conditions provided by the operation and maintenance strategy to determine the parameter settings of the inner-loop load-shedding controller. The control strategy we propose can balance the load-shedding capability of the controller with the fatigue load of the pitch actuator while considering the predefined remaining useful blade life in the operation and maintenance strategy, avoiding unplanned downtime and reducing maintenance costs.

## 1. Introduction

Wind energy plays an extremely important role in the transition to clean and sustainable energy sources. Particularly noteworthy is wind power generation, capable of large-scale production, where the significance of increasing the efficiency of power generation systems, such as through facility optimization and operation and maintenance (O&M) cost reduction, is rising as wind turbines grow in size [1,2,3]. On 16 April 2024, the Global Wind Energy Council (GWEC) released the Global Wind Energy Report 2024 in Abu Dhabi. The report showed that, in 2024, the world’s newly installed wind power capacity reached a record-breaking 117 GW, the best year in history. Figure 1 shows the top six countries with the world’s largest installed capacity as of 2024.

In Figure 1, it can be seen that China currently ranks first in the world in terms of the cumulative installed capacity of wind turbines, reaching 328.4 GW, while the United States ranks second, with a cumulative installed capacity of 132.2 GW, followed by Germany, Spain, India, and Britain.

Although the capacity of a wind turbine assembly is constantly increasing, due to frequent changes in the operating conditions of the wind turbines, as well as long-term exposure to extreme temperatures, rainfall, snow, salt spray, and complex and variable load impacts, their health status continues to deteriorate, ultimately leading to the occurrence of faults. With the continuous increase in wind turbine assembly capacity, operation and maintenance costs are gradually rising. According to the National Renewable Energy Laboratory (NREL) in the United States, the annual maintenance cost of an onshore wind turbine is approximately USD 17,000, while the maintenance cost of an offshore wind turbine is as high as USD 46,000. Therefore, adopting reasonable methods to evaluate the condition of wind turbines or predict the remaining service life of important components of the turbines is of great significance for reducing the operation and maintenance costs of wind farms and ensuring the stable operation of wind turbines. To date, the topics of evaluating the health status of wind turbines, predicting the remaining service life of important subsystems, and diagnosing faults in wind turbine units have received widespread attention from scholars.

Reference [4] proposed frequency matching demodulation transform (FMDT) to estimate the occurrence of bearing faults, which performs better under high-noise conditions compared to traditional methods. Reference [5] proposed a state space estimator for wind turbine bearing faults, which has self-constraint properties and can update the state space model in the future with good robustness. Its disadvantage is that the established state space model is relatively complex, and the accuracy of estimating state variables is relatively low. Reference [6] considered the problems of traditional time–frequency analysis in extracting the non-stationary fault frequencies of wind turbine bearings and proposes a frequency-chirprate synchro squeezing-based scaling chirplet transform (FCSSCT) method, which has much better energy concentration compared with several advanced methods. Reference [7] developed an intelligent model based on an imperceptible neural network, which can predict the remaining service life of rotating equipment without human intervention. However, this method requires a large amount of historical data to train the model, which is difficult to implement in practical engineering. Reference [8] proposed an extended network for multi-channel information fusion to predict the remaining service life of rotating equipment. This method can predict the remaining service life of equipment under different operating conditions. However, as the dimensionality of input data increases, the network structure becomes more complex, the inference time increases, and the convergence speed slows down, resulting in a decrease in the accuracy of remaining service life prediction. In addition to the research on wind turbine bearings mentioned above, many scholars have also studied methods for predicting the remaining service life of blades: for example, reference [9] used Openfast and Matlab for joint simulation to calculate the fatigue load of wind turbine blades and obtain their remaining service life. However, this method focuses more on calculating the fatigue load than on studying life prediction algorithms, and the accuracy of the service life results obtained compared to the method proposed in this paper is relatively poor. Reference [10] proposed a wind turbine blade life prediction method based on a genetic algorithm-optimized backpropagation (BP) neural network, which effectively improves the prediction accuracy of the BP neural network. However, the BP network itself has the disadvantage of slow convergence speed. When using a genetic algorithm for optimization, not only does it increase the selection of parameters, causing difficulties in parameter adjustment, but it may also amplify the slow convergence speed of the BP network. Reference [11] analyzed the blade life of two different types of small wind turbines by establishing a finite element model as a load boundary condition to predict turbine performance and reliability, thereby predicting the blade life. However, compared with the method proposed in this paper, the complexity of finite element analysis is higher.

Nowadays, advanced unit control strategies are applied to wind turbines to maximize power generation and reduce component loads. Simultaneously, reducing the load helps to extend the service life of components and allows for the establishment of larger-sized wind turbines. The active control of wind turbines is mainly achieved by controlling the electromagnetic torque and pitch angle of the wind turbine, reducing the aerodynamic load of the wind turbine and reducing component vibration. Reference [12] used an observer combined with the linear quadratic Gaussian (LQG) method to adjust the speed of a wind turbine and reduce the load on the transmission chain and blades. However, the LQG controller is complex and lacks robustness in its design. Reference [13] proposed a model predictive control (MPC) strategy for wind turbine output power fluctuations, which effectively improves the efficiency of energy conversion and reduces power fluctuations. However, when the wind speed is higher than the rated wind speed, not only should the output power of the wind turbine be considered, but the load of the unit should also be reduced. Reference [14] designed an individual pitch control (IPC) controller for a three-blade wind turbine based on Multi-Blade Coordinates (MBCs) combined with reference adaptive control to reduce the impact of unbalanced blade loads; however, this strategy may lead to frequent pitch changes and exacerbate damage to pitch actuators. To reduce the impact of process noise and measurement noise, reference [15] designed a disturbance-accommodating control (DAC) controller based on a Kalman filter to regulate rotor speed and reduce transmission chain torsional vibration. On the basis of unified pitch control, reference [16] added active damping independent pitch control and used an offline multi-objective function model to adjust PI parameters, achieving the coordinated control of wind turbine rotor speed and tower and blade vibration suppression.

However, reducing the load often accompanies a decrease in power generation and an increase in the blade pitch frequency. To balance and optimize this trade-off, appropriate weight parameters can be set. Reference [17] provides an online damage calculation model to indicate the actual health status of the blade and combines a variable-gain control strategy to balance unit power generation and blade life. This method can extend the blade life at the expense of a small amount of power. However, there is a lack of a predefined life cycle and uncertainty descriptions for future operating conditions. References [18,19], respectively, proposed Multivariate Information Perception You Look Only Once (MIP-YOLO) and a Gated Residual Network (GRN) for crack prediction in blades, which can effectively predict blade cracks. But, these studies did not investigate how to guide controller parameter adjustment based on the crack propagation results and blade life. Reference [20] effectively improved the prediction accuracy of blade crack propagation by combining hyperspectral imaging with 3D convolutional neural networks, providing an effective basis for blade fault diagnosis and life prediction. However, the weight structure of convolutional neural networks is relatively complex and requires a large amount of data for training, which is not conducive to engineering implementation. Based on the above analysis, we can find that the current methods for life prediction, both domestically and internationally, are mainly based on data-driven research. However, there are few methods that combine blade life with unit control. In fact, while conducting life prediction, adjusting the controller parameters of the unit based on the life prediction results can effectively increase the remaining life of relevant, important subsystems. That is to say, the current unit control system not only needs to ensure stable power output and reduce blade load but also needs to ensure the reliability of blade operation and the predefined service life under the requirements of operation and maintenance strategies. For this purpose, this article proposes a blade life adaptive control strategy, which cascades the life control loop onto the main controller loop. The outer loop is based on historical crack observations, and the Paris crack propagation model combined with a particle filtering algorithm is used to calculate the degradation of blade life under the crack threshold conditions provided by the operation and maintenance strategy in order to determine the parameter settings of the inner-loop load-shedding controller. The inner-loop load-shedding controller is mainly designed based on random disturbance correction control to balance the load-shedding capability of the controller and the workload of the pitch actuator while considering the predefined blade life in the operation and maintenance strategy, avoiding unplanned shutdowns, and reducing the total maintenance cost of the unit. The effectiveness of the proposed method was verified through simulation using the NREL 5 MW wind turbine model and the Open FAST-MATLAB platform (2022 version). The main contributions of this article include the following:We established a crack propagation model for blades based on Paris’ law and applied the rain-flow counting method to solve the cyclic stress on the blades in order to obtain the strength factor in the model.We propose a Bayesian-based method for predicting the remaining life of blades, which addresses the issue of infinite integration in Bayesian filters. We implement it using a particle filtering algorithm, which provides a suboptimal solution for the Bayesian estimator through Monte Carlo integration. Simulation shows that, although particle filtering has a high time cost, it has more advantages in terms of lifetime distribution and prediction stability.We designed an SDAC controller and adjusted the controller parameters based on the predicted lifespan, effectively reducing the fatigue of the pitch control actuator and extending its service life. In response to the presence of untreated process noise and measurement noise in traditional interference correction control, we have adopted the Kalman filter method to improve the robustness of the controller.We calculated the operation and maintenance costs under traditional control strategies and our proposed strategy and found that adopting the strategy proposed in this paper can effectively reduce the operation and maintenance cost of the unit, which has practical engineering significance.

This paper consists of the following five parts: Section 1 is the introduction of this paper. In Section 2, we introduce the composition of wind power generation systems, the basic wind turbine model, and traditional control strategies. In the third section, we propose a comprehensive control strategy for the lifespan of wind turbine blades that effectively utilizes the predicted remaining lifespan of the blades to guide the implementation of the unit control strategy, thereby reducing the fatigue load of the blades and extending their remaining lifespan. The fourth section of this paper is the simulation results and a discussion of the control strategy proposed in this paper, and the final section is the conclusion and outlook of the paper.

## 2. Materials and Methods

### 2.1. Wind Turbine System Model

Figure 2 shows the structural diagram of a wind power generation system. It can be seen that the wind power generation system mainly includes a wind turbine, transmission chain, generator, converter, filter circuit, transformer, etc. Among them, the control goal of the machine side is to maintain the constant output power of the unit, that is, to keep the maximum wind energy tracking below the rated wind speed and the pitch control above the rated wind speed. The main control goal of the network side is to maintain the DC bus voltage constant, and its control strategy is mainly unit power factor control. By modeling the wind turbine, we can obtain the nonlinear motion equation of the wind turbine:(1)M(q,u,t)a+f(q,v,u,t)=0

In Equation (1), ***q*** represents the displacement vector, and $q={\left[q_{1},q_{2},q_{3},q_{t},q_{g}\right]}^{T}$; *q*_1_, *q*_2_, and *q*_3_ indicate the degrees of freedom of the blade to swing; *q*_t_ indicates the degree of freedom of the tower’s front and rear bending; and *q_g_
*indicates the rotational degrees of freedom of the generator. ***v*** = *d**q***/*dt* represents the velocity vector; ***a*** = *d^2^**q***/*dt*^2^ represents the acceleration vector; ***M*** is the mass matrix; and ***f*** represents the nonlinear equation of aerodynamics. *u* is the input vector of the control system; *t* represents the current time. A linearized model can be obtained by a Taylor expansion of the nonlinear kinematic equation at the working point of *v* = 18 m/s, rotor speed *w*_r_ = 12.1 r/min, and pitch angle *β =* 14.77°.
(2)$$\left\{\begin{array}{l} \dot{x}=Ax+B_{c}u+B_{d}v\\ y=Cx \end{array}\right.$$
$$u=\left[\begin{array}{ccc} \beta_{1} & \beta_{2} & \beta_{3} \end{array}\right]$$
$$v=\left[\begin{array}{ccc} v_{1} & v_{2} & v_{3} \end{array}\right]$$
$$y={\left[\begin{array}{ccccc} {\dot{q}}_{g} & q_{1} & q_{2} & q_{3} & q_{t} \end{array}\right]}^{T}$$
$$x={\left[\begin{array}{ccccccccc} q_{t} & q_{1} & q_{2} & q_{3} & {\dot{q}}_{t} & {\dot{q}}_{g} & {\dot{q}}_{1} & {\dot{q}}_{2} & {\dot{q}}_{3} \end{array}\right]}^{T}$$

In Equation (2), *A* is the system matrix, *B_c_* is the pitch control input gain matrix, *B_d_* is the wind speed disturbance input gain matrix, *C* is the output matrix of the system, and *x* contains the state of the controlled system. *u* is the pitch vector; *v* is the wind speed vector; and *y* is the measurement value of the system.

### 2.2. The Basic Control Strategy for Wind Turbines

The relationship between the wind turbine speed and generator torque is shown in Figure 3, and its working area can be divided into the maximum wind energy tracking region, transition region, and constant-power region.

At different wind speeds, wind turbines have different control objectives, mainly controlling the pitch angle and torque. When the wind speed is high but the speed of the wind turbine is lower than the rated speed, corresponding to the A(-)B region in Figure 3, optimal torque control is adopted to maintain the optimal blade tip speed ratio and achieve maximum power point tracking. When the wind speed is above the rated wind speed, corresponding to region C(-)D in Figure 3, pitch control is executed to control the wind turbine rotor speed at the rated speed while maintaining constant torque to achieve constant power control in region C(-)D. Region B(-)C is a transitional region, where the wind turbine speed reaches the rated speed but the electromagnetic torque of the generator is still less than the rated torque. Therefore, in this region, as the wind speed increases, the electromagnetic torque of the generator is increased to achieve constant speed control.

Wind turbines mainly perform maximum power point tracking (MPPT) control below the rated wind speed, with the control objective of capturing the maximum wind energy in order to achieve the maximum economic benefit. The research and improvement of MPPT control methods [21] have always been a hot topic in the field of wind power generation. Traditional MPPT methods mainly include the optimal blade tip speed ratio method (OTSR) [22], optimal torque control (OTC) [23], and so on. According to the basic theory of wind turbines, the relationship between the wind energy utilization coefficient *Cp* and the pitch angle and blade tip speed ratio can be approximately described by the following expression:(3)$$C_{p}=(0.44-0.0167\beta )\sin[\frac{\pi (\lambda -3)}{15-0.3\beta }]-0.00184(\lambda -3)\beta$$

Figure 4 illustrates the relationship between the blade tip speed ratio and the wind energy utilization coefficient of the wind turbine at pitch angles of 0 degrees, 5 degrees, and 10 degrees. In Figure 4, we can see that the optimal blade tip speed ratio *λ_opt_* of the wind turbine corresponds to its maximum wind energy utilization coefficient *C_pmax_*. When the wind speed changes, as long as the blade tip speed ratio of the wind turbine can be maintained at the optimal *λ_opt_*, the maximum wind energy can be captured. As we know, the relationship between *λ_opt_
*and *w_opt_
*is
(4)$$w_{opt}=\frac{\lambda_{opt}v}{R}$$

As shown in Equation (4), the OTSR method usually requires calculating *w_opt_* based on the wind speed *v* and then implementing MPPT control by adjusting the speed. The control principle diagram is shown in Figure 5.

The MSC control principle in Figure 5 is shown in Figure 6.

Figure 4, Figure 5 and Figure 6 depict the typical control methods for wind turbines below the rated wind speed, while above the rated wind speed, the power of the turbine is usually maintained at a constant value by controlling the pitch angle of the turbine. It is worth noting that, when the pitch angle of the wind turbine unit changes, it will bring about changes in the aerodynamic loads on the blade. A change in wind speed leads to changes in the load of the wind turbine, especially due to the influence of wind shear and tower shadow effects, which can cause the load on the rotating surface of the wind turbine to be unbalanced. Therefore, starting from the overall perspective of wind turbines, the control of wind turbines needs to simultaneously address multiple conflicting issues, such as output power fluctuations, load reduction, and the fatigue of pitch actuators. Therefore, it can be seen that multiple-input and multiple-output controllers that handle multiple targets are very necessary.

### 2.3. Stochastic-Disturbance-Accommodating Control Based on Kalman Filter

The interference correction control theory provides a method for designing feedback controllers that can automatically detect structured interference waveforms and minimize their impact. Introducing an interference state observer into the Kalman filter and adding interference correction control input to the nominal control input can counteract the influence of interference. Therefore, for wind turbines, the blade wind speed affected by wind shear is
(5)$$v_{i}=v_{hub}{\left(1+z/h\right)}^{m}\approx v_{hub}\left(1+A_{1P}\cos\varphi_{i}\right)$$

In Equation (5), *v_i_
*is the tip wind speed of the *i*-th blade; *v_hub_* is the wind speed at the center of the hub; *z* is the relative height at the center of the wheel hub; *h* is the height of the hub center above the ground; *m* is the linear wind shear index; *φ_i_
*is the azimuth angle of the *i*-th blade; and *A*_1*p*_ is the wind shear coefficient when the wind turbine speed is doubled. According to Equations (2) and (5), the system state equation can be obtained as follows:(6)$$\left\{\begin{array}{l} \Delta \dot{x}\left(t\right)=A\Delta x\left(t\right)+B_{c}\Delta u_{c}\left(t\right)+B_{d}\Delta v\left(t\right)\\ \Delta y\left(t\right)=C\Delta x\left(t\right) \end{array}\right.$$

The effect of wind shear on blade wind speed is related to the azimuth angle of the blade. To eliminate the time-varying characteristics of the azimuth angle, coordinate transformation can be used.
(7)$$T_{MBC}=\frac{1}{3}\left[\begin{array}{ccc} \begin{array}{c} 1\\ 2\cos\varphi_{1}\\ 2\sin\varphi_{1} \end{array} & \begin{array}{c} 1\\ 2\cos\varphi_{2}\\ 2\sin\varphi_{2} \end{array} & \begin{array}{c} 1\\ 2\cos\varphi_{3}\\ 2\sin\varphi_{3} \end{array} \end{array}\right]$$

In Equation (5), the wind speed at each blade tip is affected by the speed 1P, and its variation is in the form of a cosine function. The uniform wind speed component *v_u_*, vertical shear wind speed component *v_nc_*, and horizontal shear wind speed component *v_ns_* after MBC transformation are step waveforms and do not depend on the azimuth angle. *v_ns_* can be ignored: that is, interference correction control based on MBCs is used to eliminate the interference of vu and *v_nc_*. The wind speed interference is modeled as follows:(8)$$\left\{\begin{array}{l} \Delta v^{M}\left(t\right)=\theta \Delta z_{d}\left(t\right)\\ \Delta {\dot{z}}_{d}\left(t\right)=F\Delta z_{d}\left(t\right) \end{array}\right.$$

In Equation (8), $F=\left[\begin{array}{cc} \begin{array}{c} 0\\ 0 \end{array} & \begin{array}{c} 0\\ 0 \end{array} \end{array}\right]$, $\theta =\left[\begin{array}{cc} \begin{array}{c} 1\\ 0 \end{array} & \begin{array}{c} 0\\ 1 \end{array} \end{array}\right]$, $v^{M}={\left[\begin{array}{c} v_{u}\\ v_{nc} \end{array}\right]}^{T}$, F and θ are coefficient matrixes, and $\Delta v^{M}$ and $\Delta z_{d}$ represent the disturbance of the wind speed and the disturbance of the state, respectively.

Furthermore, we can obtain a linear time-invariant model based on coordinate transformation.
(9)$$\left\{\begin{array}{l} \Delta {\dot{x}}^{M}\left(t\right)=A^{M}\Delta x^{M}\left(t\right)+B_{c}^{M}\Delta u_{c}^{M}\left(t\right)+B_{d}^{M}\Delta v^{M}\left(t\right)\\ \Delta y^{M}\left(t\right)=C^{M}\Delta x^{M}\left(t\right) \end{array}\right.$$

In Equation (9), any parameter with a superscript *M* represents the result of coordinate transformation.

The unmodeled power or model error of the wind turbine system is represented by additive uncertainty, and the system model in Equation (9) is rewritten as shown in Equation (10). Based on the basic model of the given system, the system state and the nonlinear and unmodeled dynamic effects of unknown inputs are estimated, and the matrix represents the unknown inputs acting on the known system:(10)$$\left\{\begin{array}{l} \Delta {\dot{x}}^{M}\left(t\right)=A^{M}\Delta x^{M}\left(t\right)+B_{c}^{M}\Delta u^{M}\left(t\right)+B_{d}^{M}\Delta v^{M}\left(t\right)+Nf\left(t\right)\\ \Delta y^{M}\left(t\right)=C^{M}\Delta x^{M}\left(t\right)+V\left(t\right) \end{array}\right.$$

In Equation (10), the output noise is measured, and the dynamic model assumption for the unknown input disturbance term is
(11)$$\dot{f}\left(t\right)=Df\left(t\right)+W_{f}\left(t\right)$$

In Equation (11), *D* is the Hurwitz stability matrix; *W_f_* is process interference noise. According to Equations (9)–(11), the augmented system can be obtained as follows:(12)$$\left[\begin{array}{c} \Delta {\dot{x}}^{M}\left(t\right)\\ \dot{f}\left(t\right)\\ \Delta {\dot{z}}_{d}\left(t\right) \end{array}\right]=\underset{A_{p}}{\underset{︸}{\left[\begin{array}{ccc} \begin{array}{c} A^{M}\\ 0\\ 0 \end{array} & \begin{array}{c} N\\ 0\\ 0 \end{array} & \begin{array}{c} B_{d}^{M}\theta \\ D\\ F \end{array} \end{array}\right]}}\underset{x_{p}}{\underset{︸}{\left[\begin{array}{c} \Delta x^{M}\left(t\right)\\ f\left(t\right)\\ \Delta z_{d}\left(t\right) \end{array}\right]}}+\underset{B_{p}}{\underset{︸}{\left[\begin{array}{c} B_{c}^{M}\\ 0\\ 0 \end{array}\right]}}\Delta u\left(t\right)+\underset{W_{p}}{\underset{︸}{\left[\begin{array}{c} \begin{array}{cc} 0 & 0 \end{array}\\ \begin{array}{cc} I_{r} & 0 \end{array}\\ \begin{array}{cc} 0 & I_{l} \end{array} \end{array}\right]}}W_{f}\left(t\right)$$

The output equation is
(13)$$y_{p}=\underset{Cp}{\underset{︸}{\left[\begin{array}{ccc} C & 0 & 0 \end{array}\right]}}x_{p}+V\left(t\right)$$

Although the disturbance term is unknown, *Wf(t)* and *Vt* can be assumed to have certain random characteristics, and a Kalman filter can be used to achieve optimal estimation in the feedback loop, estimating the unmeasured system state and disturbance term from the system measurement output. The observer dynamics model of an augmented system can be expressed as
(14)$$\Delta {\dot{\hat{x}}}_{p}\left(t\right)=A_{p}\Delta {\hat{x}}_{p}\left(t\right)+B_{p}\Delta u\left(t\right)+K_{p}\left(\Delta y_{p}\left(t\right)-\Delta {\hat{y}}_{p}\left(t\right)\right)$$
(15)$$\Delta {\hat{y}}_{p}=C_{p}\Delta {\hat{x}}_{p}$$

In Equations (14) and (15), any parameter with the superscript ^ represents an estimated value.

The Kalman gain $K_{p}=P_{f}C_{p}^{T}R_{f}^{-1}$, and $P_{f}$ can be obtained by solving the Riccati equation:(16)$${\dot{P}}_{f}=A_{p}P_{f}+P_{f}A^{T}-P_{f}C_{p}^{T}R_{f}^{-1}C_{p}P_{f}+W_{p}Q_{f}W_{p}^{T}$$

In Equation (16), *R*_f_ is the covariance matrix representing measurement noise, and *Q_f_
*is the covariance matrix representing process noise caused by *W_f_*. The state feedback control input of the system is as follows:(17)$$\Delta u\left(t\right)=\Delta u_{x}\left(t\right)+\Delta u_{f}\left(t\right)+\Delta u_{d}\left(t\right)$$

In Equation (17), $\Delta u\left(t\right)$ is the overall control quantity, $\Delta u_{x}\left(t\right)$ is used to achieve control objectives, $\Delta u_{f}\left(t\right)$ is used to eliminate the impact of unmodeled dynamics, and $\Delta u_{d}\left(t\right)$ is used to eliminate the influence of external disturbances *z_d_*(*t*).

Among them, the nominal control input is used to achieve control objectives obtained through standard pole configuration or the LQR method. The other inputs are used to eliminate the effects of nonlinearity and unmodeled dynamics and to eliminate interference wind effects. The comprehensive control signal is
(18)$$\Delta u\left(t\right)=\left[\begin{array}{ccc} G_{x} & G_{f} & G_{d} \end{array}\right]\left[\begin{array}{c} \Delta x^{M}\left(t\right)\\ f\left(t\right)\\ \Delta z_{d}\left(t\right) \end{array}\right]$$

The system is rewritten by combining the above equations as follows:(19)$$\Delta {\dot{x}}^{M}\left(t\right)=\left(A^{M}+B_{c}^{M}G_{x}\right)\Delta x^{M}\left(t\right)+\left(B_{d}^{M}\theta +B_{c}^{M}G_{d}\right)\Delta z_{d}\left(t\right)+\left(N+B_{c}^{M}G_{f}\right)f\left(t\right)$$

To eliminate the influence of interference terms *z_d_*(*t*) and *f*(*t*), it is necessary to minimize the norm sum, ${\left\|B_{c}^{M}G_{d}+B_{c}^{M}\theta \right\|}_{2}$ and ${\left\|N+B_{c}^{M}G_{f}\right\|}_{2}$; here, the generalized inverse is used to solve it:(20)$$\left\{\begin{array}{l} G_{d}=-{\left[{\left(B_{c}^{M}\right)}^{T}B_{c}^{M}\right]}^{-1}{\left(B_{c}^{M}\right)}^{T}B_{d}^{M}\theta \\ G_{f}=-{\left[{\left(B_{c}^{M}\right)}^{T}B_{c}^{M}\right]}^{-1}{\left(B_{c}^{M}\right)}^{T}N \end{array}\right.$$

The structure of the random disturbance correction pitch control system based on the Kalman filter is shown in Figure 7. The degree of blade load shedding and pitch frequency can be coordinated through the weight matrix ***Q*** and input matrix ***R*** in the *LQR* method [24].

### 2.4. Paris Crack Propagation Model

In order to establish a reliable damage model, it is necessary to first consider the failure mechanism in order to track its development throughout the entire blade life. However, due to the complex physical phenomena themselves and limited information on blade fatigue behavior, determining the failure mechanism of blades is not an easy task. The assumption of the damage model used in this article is that failure occurs when the crack in the adhesive joint reaches a certain length, *a_fail_*. The damage model assumes the following three stages:

Cracks appear on the blades when the fan is put into use.

During the operation of the fan, cracks propagate on the blades.

When the crack length reaches the limit value *a*_fail_, the blade experiences a major failure.

Once a crack occurs on a blade, its growth simulation can be carried out using fracture mechanics methods. It is assumed that there is a one-dimensional crack along the length of the blade. Therefore, the propagation of fatigue cracks can be calculated based on periodic stress using Paris’ law: that is, the stress time-series data at the blade node can be used to generate the crack propagation speed at the corresponding node. For the convenience of calculation, assuming the uniformity of the blade material, this article uses the fatigue crack propagation equation based on Paris’ law to calculate the one-dimensional crack at the dangerous node of the blade. The crack propagation rate is as follows:(21)$$\frac{da}{dt}=\frac{A{\left(\Delta K\right)}^{m}}{{\left(1-R\right)}^{m(1-\lambda_{\omega })}}$$

In Equation (21), *a* is the current crack length, Δ*K* is the stress intensity factor, *R* is the stress ratio, and *m* and λ*_w_* are the variable parameters that can be obtained from Table 1. The expressions for the stress intensity factor Δ*K* and stress ratio *R* in Equation (21) are as follows:(22)$$\Delta K=\sqrt{\frac{\sum_{i=1}^{N}\Delta S_{i}^{2}}{N}}\sqrt{\pi a}\alpha$$
(23)$$R=\frac{S_{\min,\mathrm{rms}}}{S_{\max,\mathrm{rms}}}$$

In Equations (22) and (23), *a* is the current crack length, *α* is a variable parameter, *ΔS* is the stress range in each stress cycle, *S*_min_,_rms_ is the root-mean-square value of the minimum stress in the stress cycle, *S*_max,rms_ is the root-mean-square value of the maximum stress in a stress cycle, and *R* is the ratio of root-mean-square stress.

The calculation of the stress intensity factor *ΔK* depends on the cyclic stress on the blade and requires the use of blade load time-series data to obtain it through the rain-flow counting (RFC) method. The blade flapping moment can be used as the main load for the model input, while the aerodynamic load is mainly affected by the control strategy, average wind speed, and turbulence intensity. *A*, *m*, *λ*, *ω*, and *α* in the above equations are all material coefficients, which can be calculated based on the blade’s specified material or selected based on the average failure rate of the actual blade. The parameter values selected in this article are shown in Table 1.

For the convenience of iterative calculation, Equation (21) can be transformed into the following discrete form using the Euler method:(24)$$a_{k}=\frac{A{\left(\Delta K\right)}^{m}}{{\left(1-R\right)}^{m\left(1-\lambda_{\omega }\right)}}dt+a_{k-1}=g\left(a_{k-1}\right)$$

The prediction problem of damage models is always affected by uncertainty. The sources of uncertainty can be the errors between the established damage model and the actual physical model, the errors in the measurement environment and the sensor itself, the randomness of future wind speed inputs, and the impact of potential faults on long-term damage models.

### 2.5. Blade Life Prediction Based on Particle Filter

Using nonlinear state transition equations through Bayesian filters [25], with *f_a_* as a function of the damage state, we defined the damage state or crack length *a_k_* at time *k*, which is as a function of the damage state *a_k_*_−1_ at time *k*−1, satisfying the first-order Markov process assumption. The measured damage state is represented by the observation model *h*, which satisfies the assumption of independent observation:(25)$$a_{k}=f_{a}\left(a_{k-1},w_{a,k}\right)$$
(26)$$y_{a}=h\left(a_{k},\nu_{a,k}\right)$$

In Equations (25) and (26), *w_a,k_* describes the non-Gaussian additive noise of the uncertainty of the damage process model; non-Gaussian additive noise of the uncertainty of measurement process models is defined by *v_ak_*. In the Bayesian prediction step, the prior probability *p*(*a_k_*|*y*_1:*k*−1_) at time *k* is estimated by combining the posterior probability *p*(*a_k_*_−1_|*y*_1:*k*−1_) at time *k*−1 with the Chapman–Kolmogorov equation, and its expression is as follows:(27)$$p\left(a_{k}\left|y_{1:k-1}\right.\right)=\int p\left(a_{k}\left|a_{1:k-1}\right.\right)p\left(a_{k-1}\left|y_{1:k-1}\right.\right)da_{k-1}$$

In the Bayesian update step, the prior probability *p*(*a_k_*|*y*_1:*k*−1_) in Equation (26) is updated to the posterior probability *p*(*a_k_*|*y*_1:*k*_) at time *k* by observing the model and the observed values at time *k*. Its expression is as follows:(28)$$p\left(a_{k}\left|y_{1:k}\right.\right)=\frac{p\left(y_{k}\left|a_{k}\right.\right)p\left(a_{k}\left|y_{1:k-1}\right.\right)}{p\left(y_{k}\left|y_{1:k-1}\right.\right)}$$

The normalization constant *p*(*y_k_*|*y*_1:*k*−1_) in Equation (27) is calculated as follows:(29)$$p\left(y_{k}\left|y_{1:k-1}\right.\right)=\int p\left(y_{k}\left|a_{k}\right.\right)p\left(a_{k}\left|y_{1:k-1}\right.\right)dx_{k}$$

The disadvantage of Bayesian filters is the problem of infinite integration. For complex nonlinear functions, analytical solutions cannot be obtained. Particle filters provide suboptimal solutions for Bayesian estimation problems through Monte Carlo integration. A set of random samples and related weights are used to approximate the probability density function, and the sample mean is used to replace the integration operation. This can effectively handle filtering problems with a nonlinear state space and non-Gaussian noise distribution. The Monte Carlo method based on the random sampling operation can transform the integration operation into a summation operation of finite sample points, and the Dirac function *δ*(*a*) is incorporated to calculate the expected approximate posterior probability *p*(*a_k_*|*y*_1_:*_k_*) of particles:(30)$$p\left(a_{k}\left|y_{1:k}\right.\right)=\sum_{i=1}^{N}w_{k}^{\left(i\right)}\delta \left(a_{k}-a_{k}^{\left(i\right)}\right)$$

In fact, the posterior probability is unknown, so the reference distribution *q*(*a_k_*|*y*_1:*k*_) is used to sample particles and calculate their weights $w_{k}^{\left(i\right)}$:(31)$$a_{k}^{\left(i\right)}∼q\left(a_{k}\left|y_{1:k}\right.\right)$$
(32)$$w_{k}^{\left(i\right)}=\frac{p\left(y_{1:k}\left|a_{k}\right.\right)p\left(a_{k}\right)}{q\left(a_{k}\left|y_{1:k}\right.\right)}$$
(33)$$q\left(a_{k}\left|y_{1:k}\right.\right)=q\left(a_{k-1}\left|y_{1:k-1}\right.\right)q\left(a_{k}\left|a_{k-1},y_{1:k}\right.\right)$$
(34)$$w_{k}^{\left(i\right)}=w_{k-1}^{\left(i\right)}\frac{p\left(y_{k}\left|a_{k}^{\left(i\right)}\right.\right)p\left(a_{k}^{\left(i\right)}\left|a_{k-1}^{\left(i\right)}\right.\right)}{q\left(a_{k}^{\left(i\right)}\left|a_{k-1}^{\left(i\right)}\right.,y_{k}\right)}$$

To solve the problem of particle iteration, sequential importance sampling is adopted to consider time-series data and improve filtering efficiency. Equations (32) and (33) are iterative calculation formulas for the reference distribution and particle weight. In a particle filter, if the reference distribution is selected as *p*(*a*^(*i*)^*_k_*|*a*^(*i*)^*_k_*_−1_), the particle sampling formula and weight iteration operation formula can be simplified as follows:(35)$$a_{k}^{\left(i\right)}∼p\left(a_{k}^{\left(i\right)}\left|a_{k-1}^{\left(i\right)}\right.\right)$$
(36)$$w_{k}^{\left(i\right)}=w_{k-1}^{\left(i\right)}p\left(y_{k}\left|a_{k}^{\left(i\right)}\right.\right)$$

Particle filtering may experience particle degradation during iterative operations, which can reduce prediction accuracy. For this reason, we designed importance resampling algorithms, which involve resampling samples, replicating particles with high retention weights, and eliminating particles with low weights to suppress degradation. Due to the independent and identically distributed new particles, the new weights are updated to
(37)$$w_{k}^{\left(i\right)}=\frac{1}{N}$$

Therefore, the state estimation after particle resampling is
(38)$${\hat{a}}_{k}\approx \sum_{i=1}^{N}a_{k}^{\left(i\right)}$$

For the observation data of blade cracks, measurement noise needs to be added to cope with the error effects encountered in sensors or inspections in practical applications. The actual measured crack propagation data are simulated, and different degrees of noise and deviation are added to the actual crack size; first, a deterministic deviation of 3 mm is added, and then the random noise is evenly distributed between −1 mm and 1 mm. The parameter distribution in Table 1 is used for the Paris model in long-term prediction to compensate for the impact of potential damage to other components on the long-term blade damage model. The final blade life prediction process based on the Bayesian model is shown in Figure 8.

The following are the steps for predicting the blade life based on Bayesian models:(1)When *k* = 1, establish a state space equation based on the damage degradation model of the blade, including the state transition equation and observation equation. According to the expected parameter initialization model in Table 1, introduce the error distribution in the iterative calculation of the long-term predicted crack propagation model, with the adjustment parameters *Q_a_*_,*k*_ and *R_a_*_,*k*_ set.(2)Perform the Bayesian prediction step based on particle filtering and include the observation equation for the Bayesian update step.(3)If the estimated state after updating exceeds the threshold, calculate the remaining service life. If it does not exceed the threshold, let the prediction time *k* = *k* + 1, go back to the second step, and continue the iteration.

The operation control strategy in this article is to use OTC to achieve maximum wind energy tracking below the rated wind speed, linear torque control in the transition zone, and stochastic-disturbance-accommodating control combined with constant torque control above the rated wind speed to achieve constant power control. We have previously conducted some research to prove that SDAC has a significant effect on load reduction, but it also adds a huge burden to the pitch actuator, requiring coordination between load reduction and vibration reduction and the pitch execution rate. Therefore, an outer-loop controller is connected to the main controller, which increases or decreases the load reduction ability of the controller according to the lifespan requirements through the weight ratio in order to change the crack propagation rate of the blades and change the lifespan distribution of each blade when it reaches the repair reference value *a_rep_* and the fault threshold thereafter, achieving an integrated control strategy for the blade lifespan. In order to achieve the repair reference value *a_rep_* at the same time for the same node of each blade, it is necessary to predict the distribution of its lifespan in advance and adjust the controller parameters in a timely manner. Here, a lifespan prediction method based on particle filtering is used to obtain the lifespan distribution of *a_ref_* at the specified reference crack, as shown in Figure 9.

According to the distribution of the lifespan, the following lifespan efficiency coefficients can be defined:(39)$$L_{Bi}=\frac{L_{\mathrm{ref}}-L_{e}}{L_{b}-L_{e}}$$

In Equation (39), *L_Bi_
*is the effective life coefficient of blade *i* at node 1; *L_ref_* is the reference time node provided by the inspection and maintenance strategy to reach the reference crack area *a_ref_*; *L_e_* is the expected time node for the particle to reach the reference crack area *a_ref_*; *L_b_
*is the optimal time node for particles to reach the reference crack area *a_ref_*; and *L_w_* is the poorer time node for particles to reach the reference crack area. Based on the life coefficient and life distribution, the following actions can be taken:(1)When *L_Bi_
*< 0, the expected *L_e_* in the predicted life distribution meets the requirement *L_ref_* given by the inspection and maintenance strategy, and there is no need to change the load-shedding capacity of blade *i*.(2)When 0 < *L_Bi_
*< 1, the expected Le of the predicted life distribution does not satisfy *L_ref_*, but the time node *L_b_
*satisfies *L_ref_*. Therefore, it is necessary to strengthen the load-shedding ability of blade *i*.(3)When *L_Bi_
*> 1, the expected *L_e_* of the predicted life distribution does not satisfy *L_ref_*, and *L_b_* also does not satisfy *L_ref_*. Therefore, the load-shedding ability of blade *i* needs to be further strengthened.

The integrated control strategy for blade health is shown in Figure 10.

The blade weight factor *w_i_* in the controller parameters is adjusted, and the corresponding state control gain is calculated, as shown in Equation (40):(40)$$\left\{\begin{array}{l} Q=diag\left(\left[\begin{array}{ccccc} 10^{-1} & w\times 10^{-2} & w\times 10^{-2} & w\times 10^{-2} & 0\;10\;0\;0\;0 \end{array}\right]\right)\\ R=diag\left(\left[\begin{array}{ccc} 15/w & 15/w & 15/w \end{array}\right]\right) \end{array}\right.$$

## 3. Results

### Simulation Results and Analysis

The operating range of the NREL 5 MW wind turbine used in this article is from *v*_cut_ = 3 m/s to the cut-off wind speed *v*_out_ = 25 m/s. The Rayleigh distribution of wind speeds at the hub height is shown in Figure 11, which determines the average wind speed of the IEC Kaimal turbulence model. The data used in this article are all from the Supervisory Control And Data Acquisition (SCADA) system of the wind turbines at the Guyuan Wind Farm in China; the turbulence intensity is set at 8%, and the average wind speed is updated every 600 s. The probability density function and probability distribution function of the Rayleigh distribution are as follows [26]:(41)$$P\left(v\right)=1-e^{\left[-\pi {\left(v/2v_{\mathrm{ave}}\right)}^{2}\right]}$$
(42)$$p\left(v\right)=\left(\pi v/2v_{\mathrm{ave}}^{2}\right)e^{\left[-\pi {\left(v/2v_{\mathrm{ave}}\right)}^{2}\right]}$$

In Equations (41) and (42), *v* represents the actual wind speed, and *v_ave_* represents the average wind speed.

Based on the above operation control strategy, a segmented operation of the entire wind speed range is carried out between the wind speed *v_cut_* and the wind speed *v_out_*. While updating the wind speed, the time-series data of the flapping bending moment on the key damage nodes of each blade are collected: that is, 30% of the blade length at and above the blade root is node 1 and node 6. The rain-flow counting method is used to calculate the load range and number of cycles, which are used as load inputs for the Paris model to calculate the crack propagation length in the current cycle. The crack propagation observations for nodes at the root and 30% of the blade length of each blade are shown in Figure 12.

For long-term damage model prediction with process noise and measurement noise, a nonlinear prediction method based on the Bayesian model is needed to correct the influence of uncertainty, as shown in Figure 13. The turbulent wind in the variable wind speed range is used as the input condition for wind speed, and the Paris crack propagation model using the parameter distribution in Table 1 is used as the long-term prediction model. The particle filter is used to adjust the parameters *Q_a,k_* and *R_a,k_
*to correct the uncertainty influence of measurement noise and model error.

When the values of each particle reach the fault threshold, the life distribution of the blade can be obtained, as shown in Figure 14, which is mainly influenced by the particle distribution. If the model error is small, a smaller *Q_a_
*and *k* can be chosen to obtain a narrower range of life distribution, which can provide a more reliable life prediction. Conversely, a larger *Q_a_* and *k* will result in a wider life distribution, making it difficult to provide effective information for maintenance decisions. Particle filtering can provide a large number of particles to fit the distribution of the blade’s end life.

This article combines the Open-FAST Simulink joint simulation platform to verify the effectiveness of the integrated control strategy for blade damage in the short term. The input wind is the IEC Kaimal wind speed model, with a turbulence intensity of 8% and an average wind speed ranging from v = 16 m/s to v = 20 m/s. The wind speed is updated every 100 s based on a uniform distribution, with a total time length of 4200 s. Under normal constant parameter operation, as shown in Figure 10, the crack propagation speed at node 1 of blade 1 and blade 3 is faster than that of blade 2, which may cause earlier unexpected failures before maintenance. Therefore, a reference crack location *a_ref_
*= 0.01010 m and a reference time node *L_ref_
*= 3700 s can be set for blade 2. As shown in Figure 15, the integrated control strategy for blade life is applied to predict the life distribution of each blade every 1000 s. The effective life coefficient is calculated to adjust the weight factor w in Equation (40) to change the crack propagation rate so that blade 1 and blade 2 can reach the reference crack area after the reference time node *L_ref_*.

As shown in Figure 16, the integrated control strategy for blade life is applied to predict the life distribution of each blade every 1000 s. The effective life coefficient is calculated to adjust the weight factor w in Equation (43) to change the crack propagation rate, so that blade 1 and blade 2 can reach the reference crack location *a_ref_* after the reference time node *L_ref_*.

Based on the above analysis, the selected leaf weight factor *w_i_* is as follows:(43)$$w_{i}=\left\{\begin{array}{l} 1,\;i=1\\ 3\;i=2\\ 5,\;i=3 \end{array}\right.$$

As shown in Figure 17 and Figure 18, the input wind is the IEC Kaimal wind speed model with a turbulence intensity of 8% and a duration of 100 s. The average wind speed ranges from v = 16 m/s to v = 20 m/s with an interval of 1 m/s. Therefore, simulations can be performed based on the distribution of the stress intensity factor coefficient *ΔK*_0_ and the corresponding pitch rate for each blade node 1 under the weight factor *w_i_*.

Figure 17a–c respectively represent the stress intensity factor under different wind speeds and weighting factors.

As the weight factor coefficient increases, it can reduce the swinging load borne by each blade, but it will also significantly increase the pitch rate. The integrated control strategy for blade life can adjust the crack propagation rate of each blade at node 1 in Figure 10, reaching the reference crack *a_ref_* at the specified reference time node *L_ref_*, as shown in Figure 19.

When combined with the 1000th second in Figure 16 and Figure 19, the life estimates *L_e_* provided by PF for blades 1 and 3 do not satisfy *L_ref_*, while *L_b_* satisfies *L_ref_*. Therefore, it is necessary to strengthen the load-shedding capacity of blades 1 and 3, and w2 is sufficient. At 2000 s, the estimated lifespan of each blade using PF, *L_e_*, satisfies *L_ref_*, so there is no need to improve the load-shedding capacity, and w1 can be used. In Figure 12, it can also be observed that the crack propagation rate significantly slows down after 1000 s and recovers at 2000 s.

In addition to the above simulations, in order to verify the practical value of the proposed strategy, some research on unit operation and maintenance cost calculations should be considered in this paper. We have compared two commonly used operation and maintenance strategies and the operation and maintenance costs of the strategy proposed in this article. The simulation results show that using the strategy proposed in this article for control can effectively reduce the operation and maintenance costs of the unit. The total costs generated during the operation of wind turbines can usually be described using Equation (44):(44)$$C_{total}(z,e,d)=C_{ins}(z,e,d)+C_{rep}(z,e,d)+C_{cor}(z,e,d)+C_{loss}(z,e,d)$$

In Equation (44), $C_{total}(z,e,d)$ represents the total cost, $C_{ins}(z,e,d)$ represents the inspection cost, $C_{rep}(z,e,d)$ represents the preventive intervention cost, $C_{cor}(z,e,d)$ represents the corrective intervention cost, and $C_{loss}(z,e,d)$ represents the cost of electricity production losses, which depends on downtime and the input wind speed. Assuming that the initial electricity price is a constant, *b_init_*, the analysis of the cycle life model ignores inflation and electricity price fluctuations in practice. The impact and benefits generated belong to the expected electricity revenue *B*, which is used here to calculate the production loss *C_loss_* caused by downtime:(45)$$C_{loss}=b_{init}P(v)T_{h,stop}$$

In Equation (45), *b_init_* represents the electricity price constant, *T_h_*_,*stop*_ is the current inspection- and repair-related downtime, and *P*(*v*) is the output power of the wind turbine. The preventive intervention cost *C_rep_* and corrective intervention cost *C_cor_* generated by each maintenance strategy need to be combined with the transportation strategy, and the downtime *T_h_*_,*stop*_ is used for calculation. According to the guidance price for offshore wind power approved by the National Development and Reform Commission of China in 2019, it has been adjusted to 0.75 CNY/*kW·h* per kilowatt hour. Based on the downtime, the cost of electricity production loss, *C_loss_*, can be calculated. Assuming that weather conditions with wind speeds less than 10 m/s can be found and that inspection and maintenance will not be interrupted by adverse weather conditions, the downtime, *T_h,stop_*, can be expressed as the time required for inspection or maintenance. The single inspection cost *C_ins_*, preventive repair cost *C_rep_*, and corrective cost *C_coren_* are calculated based on relevant information. These costs include the wind energy damage cost *C_loss_*, transportation cost *C_tran_*, and maintenance personnel’s individual cost *C_peo_.* The transportation cost *C_tran_* is calculated based on the inspection and repair time spent each time. The number of days required for transportation *T_d,stop_* can be calculated based on the amount of downtime *T_h,stop_*. The effective working duration for maintenance work is 8 h per day. The cost calculations for each item are shown in Table 2.

The maintenance methods for wind turbines usually include maintenance based on corrective repair and maintenance based on two types. The maintenance strategy based on corrective repair does not actively intervene in the crack propagation of the blades and only initiates corrective repair when *a* > *a_fall_* causes major faults. Due to the lack of inspection and preventive repair measures, the total expected maintenance cost *C_total_* only needs to calculate the transportation cost required for corrective repair, maintenance cost of operation and maintenance personnel, downtime cost, and blade replacement cost. According to our calculations, the total cost expenditure based on corrective maintenance and time-related expenses is approximately CNY 5.94 million. Our proposed strategy can not only be combined with PF prediction to help maintenance personnel effectively identify the trend of blade crack propagation, given historical observations, but also be used to carry out maintenance work in a timely manner when the blade reaches the maintenance reference value. At the same time, combined with the integrated life control strategy, it can effectively ensure that the length of each blade crack during the preventive maintenance cycle does not exceed the maintenance reference value *a_rep_* to the extent possible, preventing maintenance personnel from carrying out preventive maintenance on each blade in stages, reducing the total maintenance time required for all blades of the unit throughout the entire life cycle, and lowering the expected total cost. The following Table 3 compares the expected and total costs of the corrective maintenance strategy, the preventive maintenance strategy, and our proposed strategy. Among them, *C_loss_* is an uncertain cost that is more important in corrective maintenance and basic preventive maintenance.

## 4. Conclusions

This article mainly focuses on the research of control strategies for wind turbines. Firstly, the basic partition operation control strategy for maximum wind energy tracking and the constant power output of wind turbines is analyzed. Secondly, the application of interference correction control in the multi-objective control of wind turbines is introduced. For the process noise and measurement noise that traditional interference correction control does not handle, a random interference correction control based on the Kalman filter is proposed to improve the robustness of the controller. Simulations show that this method can further improve the control performance of the controller, and it also indicates that an improvement in the load-shedding ability will further increase the pitch rate. At the same time, research on blade life prediction and integrated control was conducted, and a Paris blade damage model was established. The uncertainty of the damage model, including process noise and measurement noise, as well as the parameter distribution of the long-term model and the operating conditions based on the Rayleigh distribution, was analyzed. Particle filtering was used to deal with the impact of uncertainty on the prediction model. Simulations showed that, although particle filtering has a high time cost, it has more advantages in life distribution and prediction stability. At the same time, particle filtering was included to verify the feasibility of the integrated control strategy for blade life based on the effective life coefficient. Our future research may include exploring the scalability of control strategies for different types of wind turbines or environmental conditions, as well as qualitative research on how to further improve life prediction capabilities. For example, integrating machine learning techniques or processing redundant data before predicting the lifespan of wind turbines can improve the prediction performance to a certain extent.

## Figures and Tables

**Figure 1 sensors-24-05729-f001:**
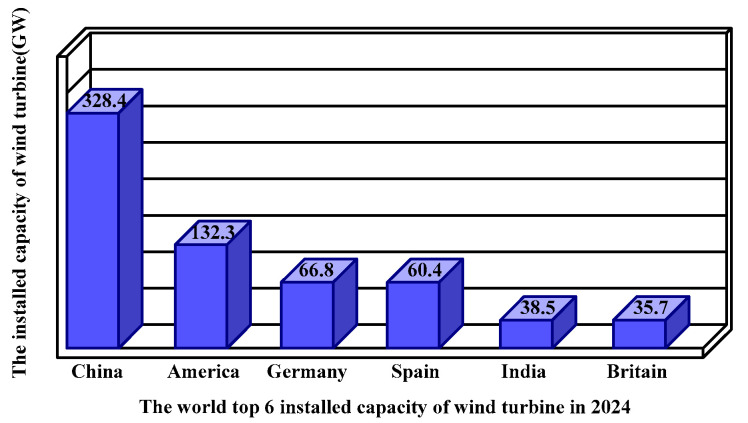
Accumulated installed capacity of various countries.

**Figure 2 sensors-24-05729-f002:**
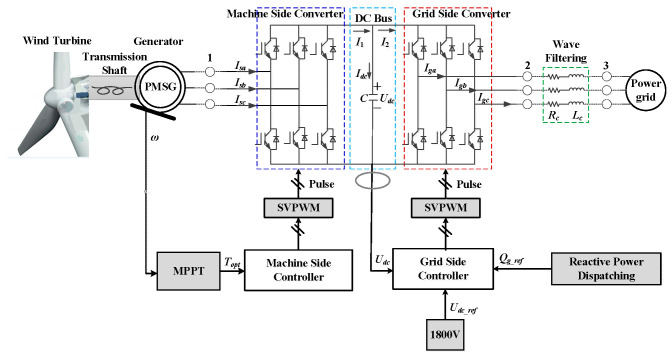
Block diagram of wind power generation system.

**Figure 3 sensors-24-05729-f003:**
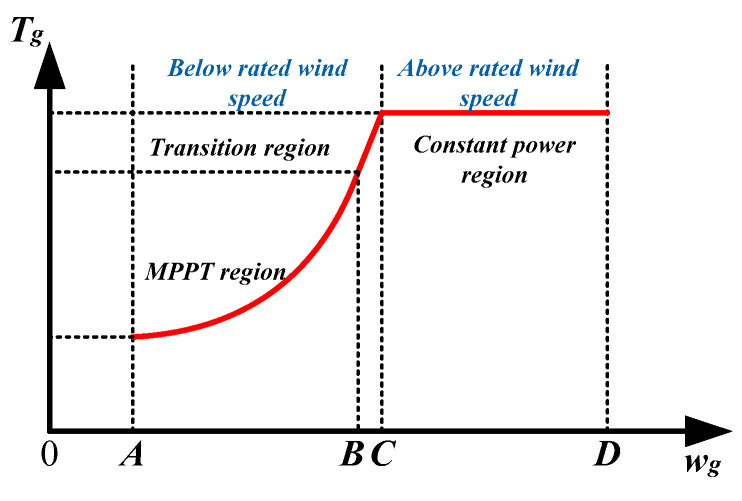
Different operating regions of wind turbine.

**Figure 4 sensors-24-05729-f004:**
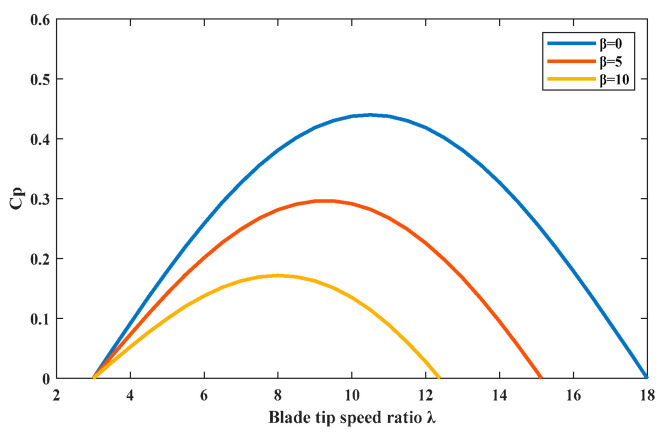
The relationship between the tip speed ratio and wind energy utilization coefficient.

**Figure 5 sensors-24-05729-f005:**
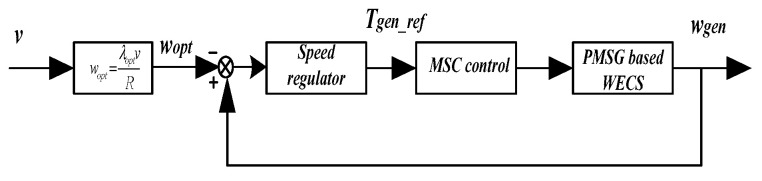
Schematic diagram of OTSR control.

**Figure 6 sensors-24-05729-f006:**
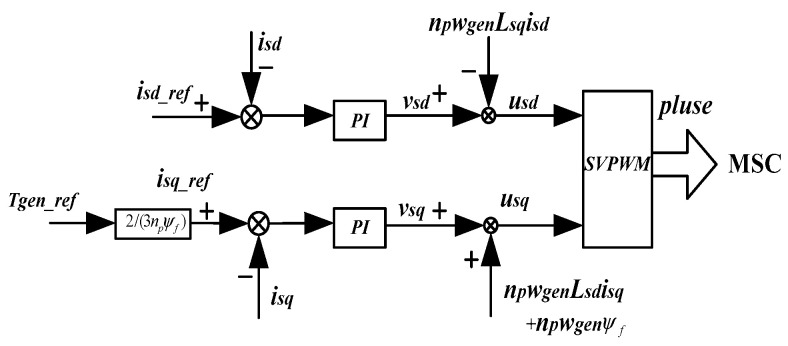
MSC control principle block diagram.

**Figure 7 sensors-24-05729-f007:**
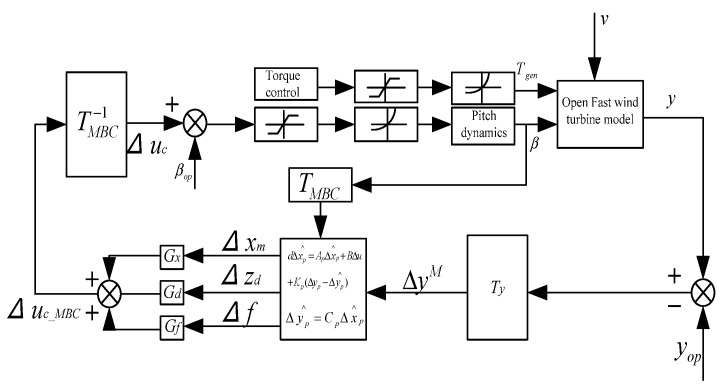
Model-free adaptive control block diagram.

**Figure 8 sensors-24-05729-f008:**
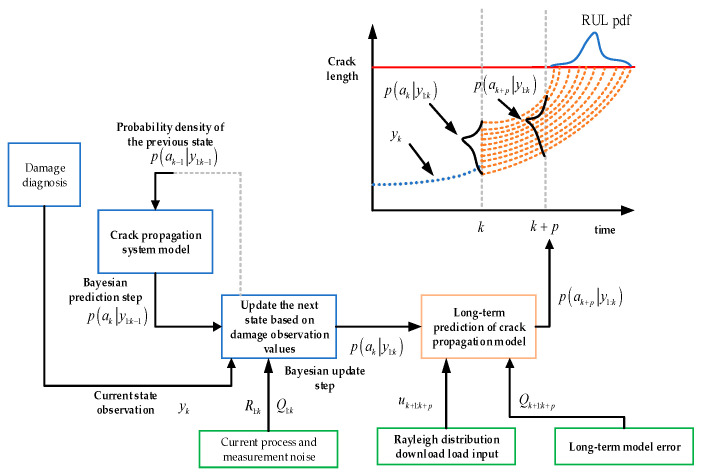
Remaining useful life prediction process based on Bayesian method.

**Figure 9 sensors-24-05729-f009:**
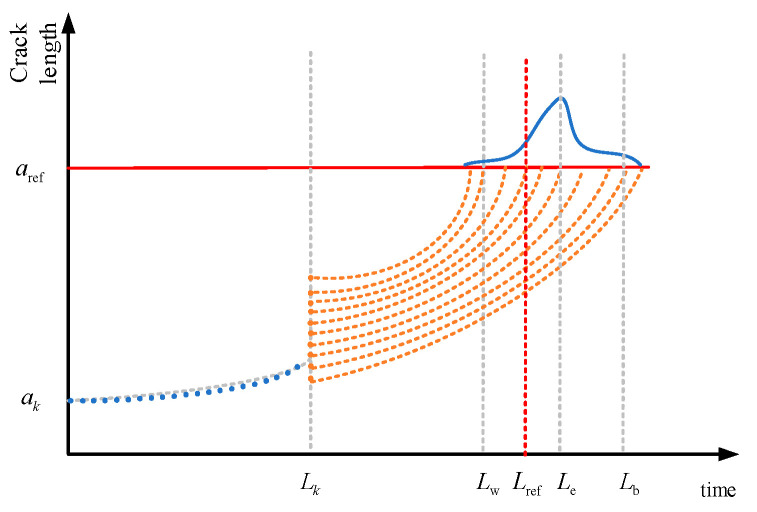
Blade life prediction based on particle filter.

**Figure 10 sensors-24-05729-f010:**
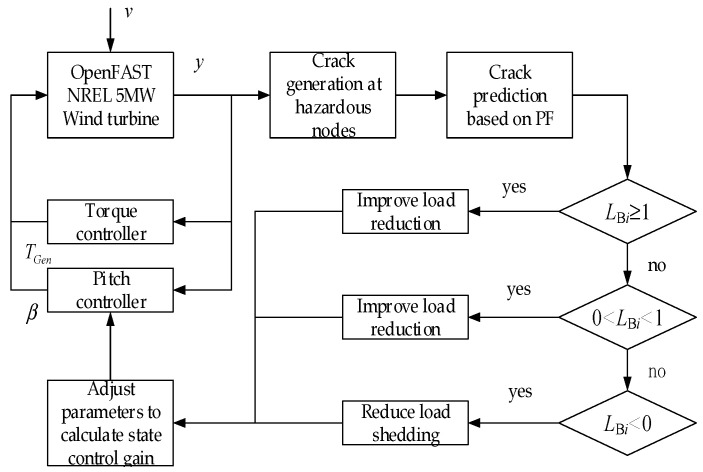
Integrated control strategy for blade life.

**Figure 11 sensors-24-05729-f011:**
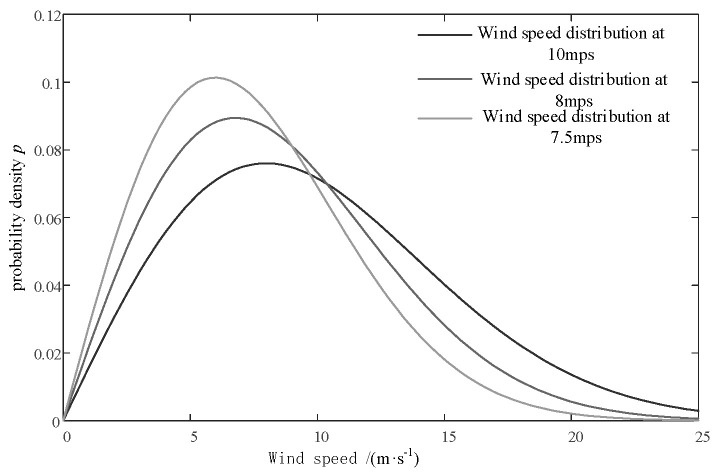
Rayleigh distribution of wind speeds for different annual wind speeds.

**Figure 12 sensors-24-05729-f012:**
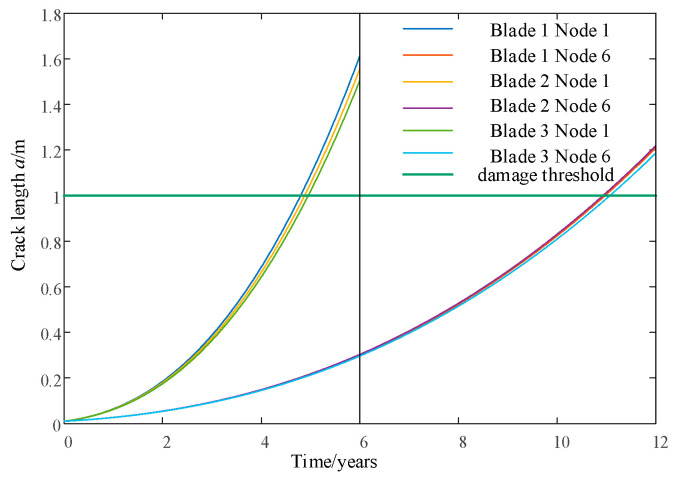
Comparison of crack observations at constant and variable wind speeds.

**Figure 13 sensors-24-05729-f013:**
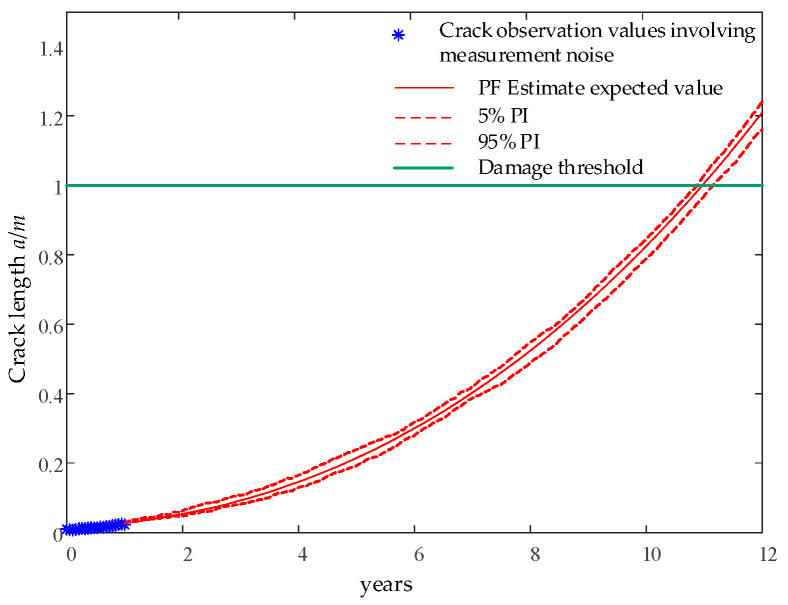
Prediction of blade 1, node 6 crack based on particle filter.

**Figure 14 sensors-24-05729-f014:**
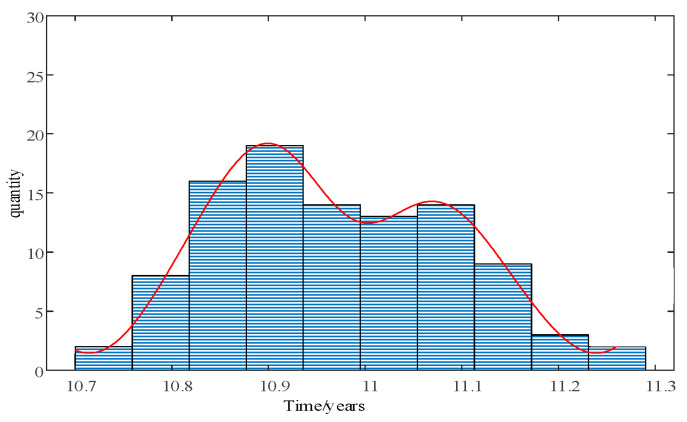
Particle-filter-based blade node 6 end-life distribution.

**Figure 15 sensors-24-05729-f015:**
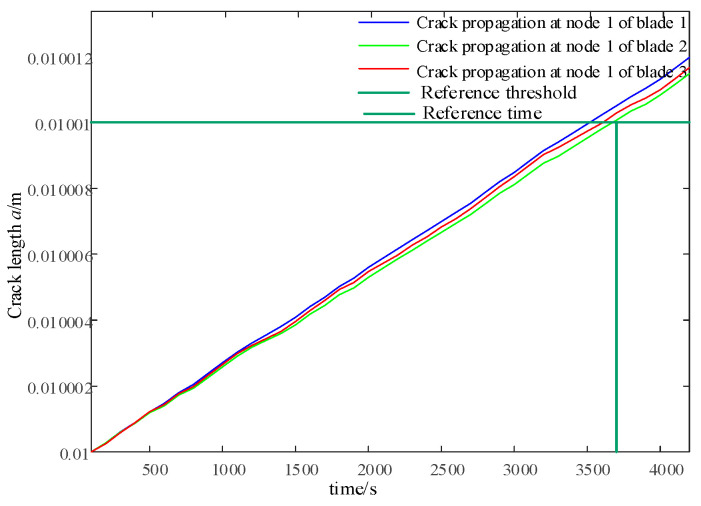
Crack observation factor with constant weight.

**Figure 16 sensors-24-05729-f016:**
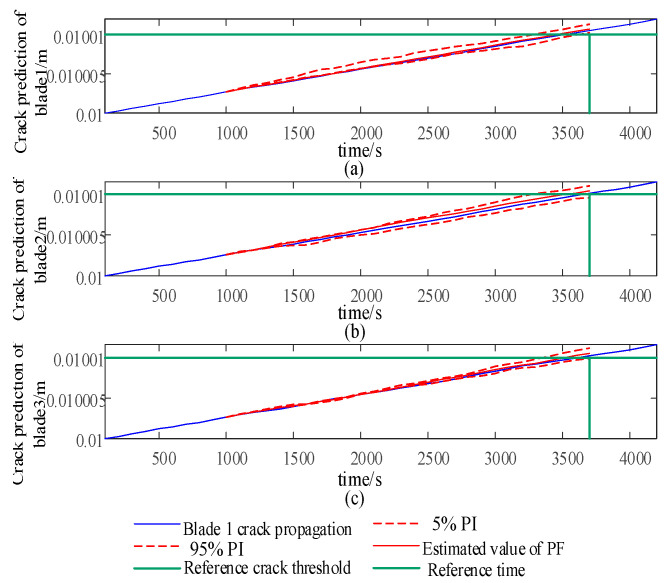
Crack prediction of blade node 1 at 1000 s (**a**–**c**).

**Figure 17 sensors-24-05729-f017:**
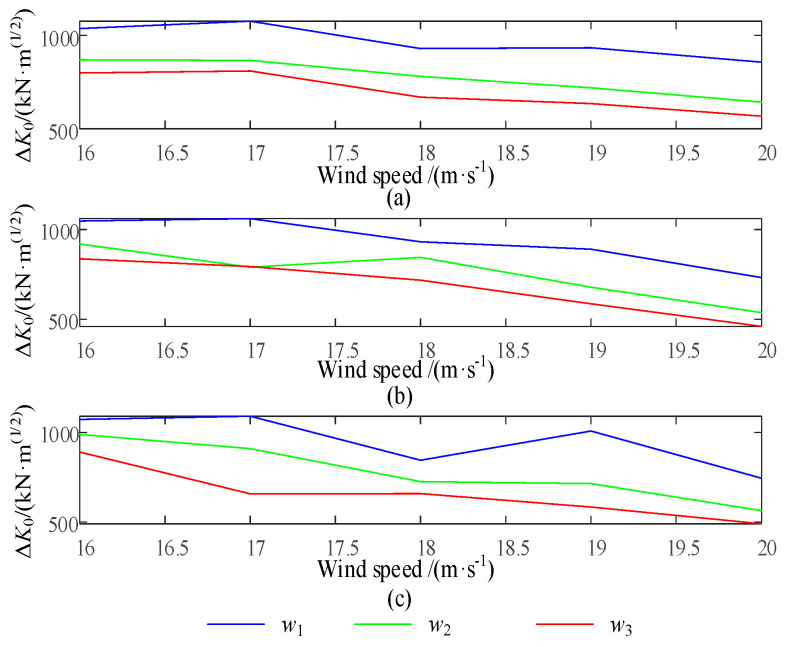
Stress intensity coefficient of each blade for different wind speeds with varying factors.

**Figure 18 sensors-24-05729-f018:**
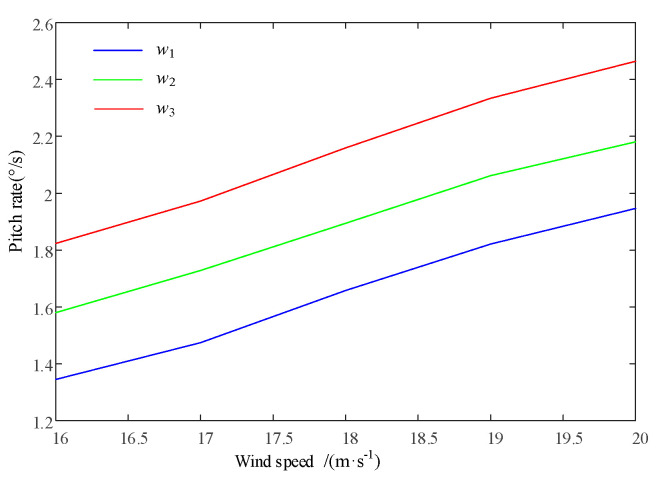
Variation in pitch speed at different wind speeds based on variable load-shedding weight factors.

**Figure 19 sensors-24-05729-f019:**
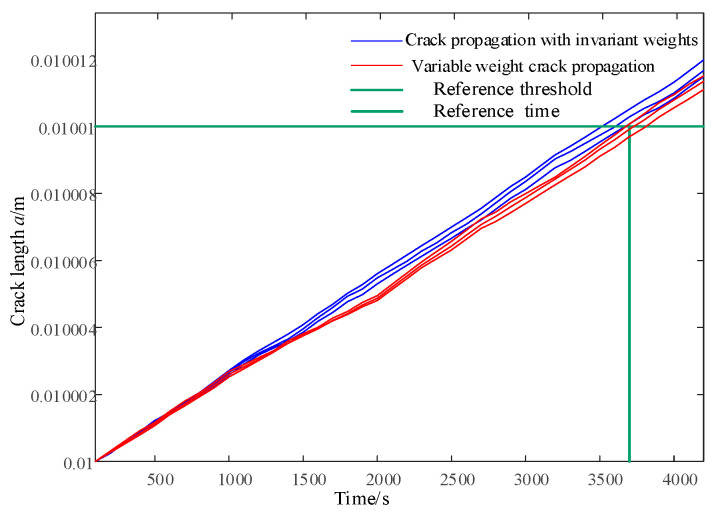
Pitch angle considering actuator delay.

**Table 1 sensors-24-05729-t001:** Crack growth model parameters.

Para.	Expectations	Variance	Distribution
*A*	1.2 × 10^−9^	0.05	*N*
*m*	1.2	0.1	*N*
*λ_ω_*	0.8	0.01	*N*
*α*	1.0	-	-
*a*	-	5 × 10^−3^	*N*

**Table 2 sensors-24-05729-t002:** Calculation of maintenance cost for a single wind turbine.

Costs	Calculation Results
Wind energy loss cost *C_loss_*, CNY	0.75×*T_h_*_,*stop*_×*P(v)*
Transportation cost *C_tran_*, CNY	50,000×*T_d_*_,*stop*_
Cost for operation and maintenance personnel *C_peo_*	300×*T_d_*_,*stop*_
Inspection cost *C_ins_*, CNY/time	*C_loss_ + C_tran_ *+ 3×*C_peo_*
Preventive repair cost of leaves *C_rep_*, CNY/time	63,200 + *C_loss_ *+ *C_tran_ *+ 3×*C_peo_*
Leaf corrective repair cost *C_cor_*, CNY/time	63,200 + *C_loss_ *+ *C_tran_ *+ 12×*C_peo_*

**Table 3 sensors-24-05729-t003:** Maintenance costs using different strategies.

Maintenance Strategy	Inspection Cost	Preventive Costs	Corrective Costs	Expected Total Cost
Corrective Maintenance	0	0	*C_loss_ *+ 4.26 × 10^6^	*C_loss_ *+ 4.26 × 10^6^
Defensive Maintenance	*C_loss_ *+ 0.572 × 10^6^	*C_loss_ *+ 1.5984 × 10^6^	0	*C_loss_ *+ 2.17 × 10^6^
Our strategy	*C_loss_ *+ 0.572 × 10^6^	*C_loss_ *+ 0.912 × 10^6^	0	*C_loss_ *+ 1.484 × 10^6^

## Data Availability

We are willing to provide the dataset for this study.

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
