# Peer review of "Research on Integrated Control Strategy for Wind Turbine Blade Life"

_sensors, 2024, doi:10.3390/s24175729_

Round 1
Reviewer 1 Report
Comments and Suggestions for Authors
This work is interesting and well executed.
Author Response
Comments1: This work is interesting and well executed.
Response1:
Thank you very much for taking the time to review this manuscript. We appreciate your recognition of our research work and wish you all the best in your work and life.

Reviewer 2 Report
Comments and Suggestions for Authors
For the process noise and measurement noise that traditional interference correction control does not address, a random interference correction control based on Kalman filter is proposed to improve the robustness of the controller. The experimental analysis verifies the effectiveness of the proposed method. The structure of the article is clear and the proposed method is novel. To further improve the quality of the paper, the following suggestions should be considered:
1. The practical value of this research work should be clarified and highlighted in the Abstract, which can help readers understand the engineering background of this research work.
2. The full name of a professional word should be given the first time it appears in the main body of the paper.
3. In introduction, more wind turbine related references should be discussed, such as bearing weak fault feature extraction under time-varying speed conditions based on frequency matching demodulation transform, and frequency-chirprate synchrosqueezing-based scaling chirplet transform for wind turbine nonstationary fault feature time–frequency representation.
4. The authors should summarize the contributions of the proposed method in the introduction.
5. The description for parameters in each equation should be added.
6. There are some grammar errors in this manuscript. Please check the whole manuscript and address these kinds of issues throughout it.
7. It is suggested that some recommendations for future work be included at the end of the Conclusion.
Comments on the Quality of English LanguageModerate editing of English language required.
Author Response
Comments 1: The practical value of this research work should be clarified and highlighted in the Abstract, which can help readers understand the engineering background of this research work. |
Response 1: Thank you for pointing this out. We agree with this comment. Therefore, we have made revisions to the abstract of the paper based on your review comments. The specific changes are as follows:
Wind turbine blades bear the maximum cyclic load and varying self weight in turbulent wind environment, which accelerates the propagation of cracks and ultimately progresses from minor faults, resulting in blade failure and significant maintenance and shutdown costs. To address the above issue, this paper proposes an adaptive control strategy for blade lifespan based on Bayesian theory. The control system is divided into internal control loop and external control loop. The outer loop is based on the Pair crack propagation model combined with particle filtering algorithm to calculate the degradation of blade life under the crack threshold conditions provided by the operation and maintenance strategy, in order to determine the parameter settings of the inner loop load shedding controller. The control strategy we propose can balance the load rejection capability of the controller and the fatigue load of the pitch actuator, while meeting the predefined remaining service life of the blades for operation and maintenance strategies, avoiding unplanned shutdowns, reducing maintenance costs, and has engineering application value.
|
Comments 2: The full name of a professional word should be given the first time it appears in the main body of the paper. |
Response 2: We have checked the professional vocabulary in the article and provided its full name when it first appeared. The specific changes are as follows: Reference [4] proposes frequency matching demodu-lation transform(FMDT) to estimate the occurrence of bearing faults, which performs better under high noise conditions compared to traditional methods. Reference [6] considers the problems of traditional time-frequency analysis in extracting non-stationary fault frequencies of wind turbine bearings and proposes a frequency-chirprate synchrosqueezing-based scaling chirplet transform(FCSSCT) method, which has much better energy concentration compared with several advanced methods. Reference [10] proposed a wind turbine blade life prediction method based on genetic algorithm optimized back propagation (BP) neural network, which effectively improves the prediction accuracy of BP neural network. Reference [12] uses an observer combined with the linear quadratic gaussian (LQG) method to adjust the speed of a wind turbine and reduce the load on the transmission chain and blades; However, the LQG controller is complex and lacks robustness in design. Reference [13] proposes a model predictive control (MPC)strategy for wind turbine output power fluctuations, which effectively improves the efficiency of energy conversion and reduces power fluctuations. However, when the wind speed is higher than the rated wind speed, not only should the output power of the wind turbine be considered, but also the load of the unit should be reduced; reference [14] designed an individual pitch control (IPC) controller for a three blade wind turbine based on Multi Blade Coordinate (MBC) combined with reference adaptive control to reduce the impact of blade unbalanced loads; However, this strategy may lead to frequent pitch changes and exacerbate damage to pitch actuators. To reduce the impact of process noise and measurement noise. Reference [15] designs an disturbance accommodating control (DAC) controller based on a Kalman filter to regulate rotor speed and reduce transmission chain torsional vibration. Wind turbine mainly perform maximum power point tracking (MPPT) control below the rated wind speed, with the control objective of capturing the maximum wind energy in order to achieve the maximum economic benefits. Reference [16] added active damping independent pitch control, and used an offline multi-objective function model to adjust PI parameters, achieving coordinated control of wind turbine rotor speed, tower and blade vibration suppression. However, reducing the load often accompanies a decrease in power generation and an increase in blade pitch frequency. To balance and optimize this trade-off, setting appropriate weight parameters can be used. Reference [17] provides an online damage calculation model to indicate the actual health status of the blade, and combines variable gain control strategy to balance unit power generation and blade life. This method can extend blade life at the expense of a small amount of power. However, there is a lack of predefined life cycle and uncertainty descriptions for future operating condition. References [18] and [19] respectively proposed Multivariate Information Perception You Look Only Once (MIP-YOLO) and Gated Residual Network (GRN) for crack prediction of blades, which can effectively predict blade cracks. But these studies did not investigate how to guide controller parameter adjustment based on crack propagation results and blade life. Reference [20] effectively improved the prediction accuracy of blade crack propagation by combining hyperspectral imaging with 3D convolutional neural networks, providing effective basis for blade fault diagnosis and life prediction. However, the weight structure of convolutional neural networks is relatively complex and requires a large amount of data for training, which is not conducive to engineering implementation.
Comments 3: In introduction, more wind turbine related references should be discussed, such as bearing weak fault feature extraction under time-varying speed conditions based on frequency matching demodulation transform, and frequency-chirprate synchro squeezing-based scaling chirplet transform for wind turbine nonstationary fault feature time–frequency representation. Response 3: We have revised the introduction section of the paper according to your review comments, added references, and carefully read and cited the references you recommended. The specific modifications are as follows:
Reference [4] proposes frequency matching demodu-lation transform(FMDT) to estimate the occurrence of bearing faults, which performs better under high noise conditions compared to traditional methods. Reference [5] proposes a state space estimator for wind turbine bearing faults, which has self constraint properties and can update the state space model in the future with good robustness. Its disadvantage is that the established state space model is relatively complex, and the accuracy of estimating state variables is relatively low. Reference [6] considers the problems of traditional time-frequency analysis in extracting non-stationary fault frequencies of wind turbine bearings and proposes a frequency-chirprate synchrosqueezing-based scaling chirplet transform (FCSSCT) method, which has much better energy concentration compared with several advanced methods. Reference [7] developed an intelligent model based on an imperceptible neural network, which can predict the remaining service life of rotating equipment without human intervention. However, this method requires a large amount of historical data to train the model, which is difficult to implement in practical engineering. Reference [8] proposed an extended network for multi-channel information fusion to predict the remaining service life of rotating equipment. This method can predict the remaining service life of equipment under different operating conditions. However, as the dimensionality of input data increases, the network structure becomes more complex, inference time increases, and convergence speed slows down, resulting in a decrease in the accuracy of remaining service life prediction. In addition to the research on wind turbine bearings mentioned above, many scholars have also studied methods for predicting the remaining service life of blades, such as: reference [9] used Openfast and Matlab for joint simulation to calculate the fatigue load of wind turbine blades and obtain their remaining service life. However, this method focuses more on calculating the fatigue load rather than studying life prediction algorithms, and the accuracy of the service life results obtained compared to the method proposed in this paper is relatively poor. Reference [10] proposed a wind turbine blade life prediction method based on genetic algorithm optimized back propagation (BP) neural network, which effectively improves the prediction accuracy of BP neural network. However, the BP network itself has the disadvantage of slow convergence speed. When using genetic algorithm for optimization, not only does it increase the selection of parameters, causing difficulties in parameter adjustment, but it may also amplify the slow convergence speed of the BP network. Reference [11] analyzed the blade life of two different types of small wind turbines by establishing a finite element model as a load boundary condition to predict turbine performance and reliability, thereby predicting blade life. However, compared with the method proposed in this paper, the complexity of finite element analysis is higher. Nowadays, advanced unit control strategies are applied to wind turbines to maximize power generation and reduce component loads. Simultaneously reducing the load helps to extend the service life of components and allows for the establishment of larger sized wind turbines. Active control of wind turbines is mainly achieved by controlling the electromagnetic torque and pitch angle of the wind turbine, reducing the aerodynamic load of the wind turbine and reducing component vibration. Reference [12] uses an observer combined with the linear quadratic gaussian (LQG) method to adjust the speed of a wind turbine and reduce the load on the transmission chain and blades; However, the LQG controller is complex and lacks robustness in design. Reference [13] proposes a model predictive control (MPC)strategy for wind turbine output power fluctuations, which effectively improves the efficiency of energy conversion and reduces power fluctuations. However, when the wind speed is higher than the rated wind speed, not only should the output power of the wind turbine be considered, but also the load of the unit should be reduced; reference [14] designed an individual pitch control (IPC) controller for a three blade wind turbine based on Multi Blade Coordinate (MBC) combined with reference adaptive control to reduce the impact of blade unbalanced loads; However, this strategy may lead to frequent pitch changes and exacerbate damage to pitch actuators. To reduce the impact of process noise and measurement noise. Reference [15] designs an disturbance accommodating control (DAC) controller based on a Kalman filter to regulate rotor speed and reduce transmission chain torsional vibration. On the basis of unified pitch control, reference [16] added active damping independent pitch control, and used an offline multi-objective function model to adjust PI parameters, achieving coordinated control of wind turbine rotor speed, tower and blade vibration suppression. However, reducing the load often accompanies a decrease in power generation and an increase in blade pitch frequency. To balance and optimize this trade-off, setting appropriate weight parameters can be used. Reference [17] provides an online damage calculation model to indicate the actual health status of the blade, and combines variable gain control strategy to balance unit power generation and blade life. This method can extend blade life at the expense of a small amount of power. However, there is a lack of predefined life cycle and uncertainty descriptions for future operating conditions. References [18] and [19] respectively proposed Multivariate Information Perception You Look Only Once (MIP-YOLO) and Gated Residual Network (GRN) for crack prediction of blades, which can effectively predict blade cracks. But these studies did not investigate how to guide controller parameter adjustment based on crack propagation results and blade life. Reference [20] effectively improved the prediction accuracy of blade crack propagation by combining hyperspectral imaging with 3D convolutional neural networks, providing effective basis for blade fault diagnosis and life prediction. However, the weight structure of convolutional neural networks is relatively complex and requires a large amount of data for training, which is not conducive to engineering implementation.
Comments 4: The authors should summarize the contributions of the proposed method in the introduction. Response 4: Thank you very much for your review comments. We have summarized the contribution of the method proposed in this paper in the penultimate paragraph of the introduction. The specific content is as follows: The main contributions of this article include: 1) We propose a Bayesian based method for predicting the remaining life of wind turbine blades. 2) By using particle filtering to deal with the impact of uncertainty on prediction models, simulations have shown that although particle filtering has a high time cost, it has more advantages in terms of lifespan distribution and prediction stability. 3) Based on the prediction of the remaining service life of the blade, the parameters of the load reduction controller were adjusted, effectively reducing the fatigue load of the variable pitch actuator. Comments 5: The description for parameters in each equation should be added. Response 5: We greatly appreciate your careful and conscientious review. We have carefully checked the manuscript again and expressed the parameters of the equations in the manuscript. The specific content is as follows:
In equation (8), and are coefficient matrixs, and representing the disturbance of wind speed and the disturbance of state respectively.
In equation (9), any parameter with a superscript M represents the result of coordinate transformation. In equation(14),(15) any parameter with a superscript ^ represents an estimated value. In equation (16), Rf is the covariance matrix representing measurement noise, Qf is the covariance matrix representing process noise caused by Wf. In equation (17), is the overall control quantity, is used to achieve control objectives, is used to eliminate the impact of unmodeled dynamics, is used to eliminate the influence of external disturbances zd(t). In the formula (21): a is the current crack length, Δ K is the stress intensity factor, R is the stress ratio, m and λw are the variable parameters which can be obtained by tableâ… . In the formula (22) and (23):a is the current crack length,α is a variable parameter, Δ S is the stress range in each stress cycle; Smin,rms is the root mean square value of the minimum stress in the stress cycle; Smax,rms is the root mean square value of the maximum stress in a stress cycle, R is the ratio of root mean square stress. In fact, the posterior probability is unknown, so the reference distribution q(ak | y1: k) is used to sample particles and calculate their weights : In equation (41)and (42), v represents actual wind speed, vave represents average wind speed.
Comments 6: There are some grammar errors in this manuscript. Please check the whole manuscript and address these kinds of issues throughout it. Response 6: Thank you for pointing this out. This article has been submitted to my doctoral English supervisor for grammar check, and we have also made some language modifications based on his suggestions.
Comments 7: It is suggested that some recommendations for future work be included at the end of the Conclusion. Response 7: Suggestions for future work are essential for this paper, and we fully agree with your review comments. After the conclusion of the paper, we mentioned future research work, which includes:
Our future research may include exploring the scalability of control strategies for different types of wind turbines or environmental conditions, as well as qualitative research on how to further improve life prediction capabilities. For example, integrating machine learning techniques or processing redundant data before predicting the lifespan of wind turbines can improve prediction performance to a certain extent.
|

Reviewer 3 Report
Comments and Suggestions for Authors
Please improve the format, such as the subtitle, the font size in the figure.
Please give some description and data from the real field.
Comments on the Quality of English LanguagePlease improve the English.
Author Response
Comments 1: Please improve the format, such as the subtitle, the font size in the figure. |
Response 1: Thank you for your careful review, Thank you for your careful review. We have revised the format and charts of the paper according to your review comments to provide readers with a better reading experience. Thank you again for your suggestion.
|
Comments 2: Please give some description and data from the real field. Response 2: After reading your review comments, we realized that the data used should be described in the paper. Therefore, we have added relevant descriptions, including: The operating range of the NREL 5MW wind turbine used in this article is from vcut=3m/s to the cut-off wind speed vout=25m/s. The Rayleigh distribution of wind speed at the hub height is generally shown in Figure 13, which determines the average wind speed of the IEC Kaimal turbulence model. The data used in this article are all from the Supervisory Control And Data Acquisition (SCADA ) system of the wind turbines at the Guyuan Wind Farm in China, the turbulence intensity is set at 8%, and the average wind speed is updated every 600 seconds. We sincerely appreciate your review comments and have submitted the data used as an attachment. |

Reviewer 4 Report
Comments and Suggestions for Authors
The topic is highly relevant given the increasing reliance on wind energy and the need for efficient maintenance strategies. Addressing blade life and failure is critical for reducing operational costs and enhancing the reliability of wind turbines. The proposed adaptive control strategy, particularly the integration of the Pairs crack propagation model with a particle filtering algorithm, demonstrates a novel approach to managing blade degradation. This combination could potentially yield more accurate predictions of blade life. However, some comments should be considered:
1) The abstract is clear and well-structured, providing a concise overview of the problem, proposed solution, and expected outcomes. However, a brief mention of the methodology or experimental validation in the abstract could strengthen the overall presentation.
2) The introduction does not provide sufficient background and include all relevant references. It should be added more advanced work to show the motivation of this work. Moreover, it would be helpful to include a discussion on how this work compares to existing strategies in the literature.
3)The emphasis on reducing unplanned downtime and maintenance costs is significant. It would be beneficial to include case studies or simulations that demonstrate the effectiveness of the proposed strategy in real-world scenarios.
4) Suggestions for future research could include exploring the scalability of the control strategy to different types of wind turbines or environmental conditions. Additionally, investigating the integration of machine learning techniques could further enhance predictive capabilities.
Author Response
Comments 1: The abstract is clear and well-structured, providing a concise overview of the problem, proposed solution, and expected outcomes. However, a brief mention of the methodology or experimental validation in the abstract could strengthen the overall presentation. |
Response 1: Thank you for pointing this out. We have made revisions to the abstract of the paper according to your feedback, specifically as follows: Wind turbine blades bear the maximum cyclic load and varying self weight in turbulent wind environment, which accelerates the propagation of cracks and ultimately progresses from minor faults, resulting in blade failure and significant maintenance and shutdown costs. To address the above issue, this paper proposes an adaptive control strategy for blade lifespan based on Bayesian theory. The control system is divided into internal control loop and external control loop. The outer loop is based on the Pair crack propagation model combined with particle filtering algorithm to calculate the degradation of blade life under the crack threshold conditions provided by the operation and maintenance strategy, in order to determine the parameter settings of the inner loop load shedding controller. The control strategy we propose can balance the load rejection capability of the controller and the fatigue load of the pitch actuator, while meeting the predefined remaining service life of the blades for operation and maintenance strategies, avoiding unplanned shutdowns, reducing maintenance costs, and has engineering application value.
|
Comments 2: The introduction does not provide sufficient background and include all relevant references. It should be added more advanced work to show the motivation of this work. Moreover, it would be helpful to include a discussion on how this work compares to existing strategies in the literature. Response 2: We have added new references in the introduction, mainly including references for predicting the remaining life of important subsystems of wind turbines and references for load suppression strategies of wind turbines, both of which are relevant to this study. We have compared some of these references, and the specific content is as follows:
Reference [4] proposes frequency matching demodu-lation transform(FMDT) to estimate the occurrence of bearing faults, which performs better under high noise conditions compared to traditional methods. Reference [5] proposes a state space estimator for wind turbine bearing faults, which has self constraint properties and can update the state space model in the future with good robustness. Its disadvantage is that the established state space model is relatively complex, and the accuracy of estimating state variables is relatively low. Reference [6] considers the problems of traditional time-frequency analysis in extracting non-stationary fault frequencies of wind turbine bearings and proposes a frequency-chirprate synchrosqueezing-based scaling chirplet transform (FCSSCT) method, which has much better energy concentration compared with several advanced methods. Reference [7] developed an intelligent model based on an imperceptible neural network, which can predict the remaining service life of rotating equipment without human intervention. However, this method requires a large amount of historical data to train the model, which is difficult to implement in practical engineering. Reference [8] proposed an extended network for multi-channel information fusion to predict the remaining service life of rotating equipment. This method can predict the remaining service life of equipment under different operating conditions. However, as the dimensionality of input data increases, the network structure becomes more complex, inference time increases, and convergence speed slows down, resulting in a decrease in the accuracy of remaining service life prediction. In addition to the research on wind turbine bearings mentioned above, many scholars have also studied methods for predicting the remaining service life of blades, such as: reference [9] used Openfast and Matlab for joint simulation to calculate the fatigue load of wind turbine blades and obtain their remaining service life. However, this method focuses more on calculating the fatigue load rather than studying life prediction algorithms, and the accuracy of the service life results obtained compared to the method proposed in this paper is relatively poor. Reference [10] proposed a wind turbine blade life prediction method based on genetic algorithm optimized back propagation (BP) neural network, which effectively improves the prediction accuracy of BP neural network. However, the BP network itself has the disadvantage of slow convergence speed. When using genetic algorithm for optimization, not only does it increase the selection of parameters, causing difficulties in parameter adjustment, but it may also amplify the slow convergence speed of the BP network. Reference [11] analyzed the blade life of two different types of small wind turbines by establishing a finite element model as a load boundary condition to predict turbine performance and reliability, thereby predicting blade life. However, compared with the method proposed in this paper, the complexity of finite element analysis is higher. Nowadays, advanced unit control strategies are applied to wind turbines to maximize power generation and reduce component loads. Simultaneously reducing the load helps to extend the service life of components and allows for the establishment of larger sized wind turbines. Active control of wind turbines is mainly achieved by controlling the electromagnetic torque and pitch angle of the wind turbine, reducing the aerodynamic load of the wind turbine and reducing component vibration. Reference [12] uses an observer combined with the linear quadratic gaussian (LQG) method to adjust the speed of a wind turbine and reduce the load on the transmission chain and blades; However, the LQG controller is complex and lacks robustness in design. Reference [13] proposes a model predictive control (MPC)strategy for wind turbine output power fluctuations, which effectively improves the efficiency of energy conversion and reduces power fluctuations. However, when the wind speed is higher than the rated wind speed, not only should the output power of the wind turbine be considered, but also the load of the unit should be reduced; reference [14] designed an individual pitch control (IPC) controller for a three blade wind turbine based on Multi Blade Coordinate (MBC) combined with reference adaptive control to reduce the impact of blade unbalanced loads; However, this strategy may lead to frequent pitch changes and exacerbate damage to pitch actuators. To reduce the impact of process noise and measurement noise. Reference [15] designs an disturbance accommodating control (DAC) controller based on a Kalman filter to regulate rotor speed and reduce transmission chain torsional vibration. On the basis of unified pitch control, reference [16] added active damping independent pitch control, and used an offline multi-objective function model to adjust PI parameters, achieving coordinated control of wind turbine rotor speed, tower and blade vibration suppression. However, reducing the load often accompanies a decrease in power generation and an increase in blade pitch frequency. To balance and optimize this trade-off, setting appropriate weight parameters can be used. Reference [17] provides an online damage calculation model to indicate the actual health status of the blade, and combines variable gain control strategy to balance unit power generation and blade life. This method can extend blade life at the expense of a small amount of power. However, there is a lack of predefined life cycle and uncertainty descriptions for future operating conditions.References [18] and [19] respectively proposed Multivariate Information Perception You Look Only Once (MIP-YOLO) and Gated Residual Network (GRN) for crack prediction of blades, which can effectively predict blade cracks. But these studies did not investigate how to guide controller parameter adjustment based on crack propagation results and blade life. Reference [20] effectively improved the prediction accuracy of blade crack propagation by combining hyperspectral imaging with 3D convolutional neural networks, providing effective basis for blade fault diagnosis and life prediction. However, the weight structure of convolutional neural networks is relatively complex and requires a large amount of data for training, which is not conducive to engineering implementation.
Comments 3: The emphasis on reducing unplanned downtime and maintenance costs is significant. It would be beneficial to include case studies or simulations that demonstrate the effectiveness of the proposed strategy in real-world scenarios. Response 3: Thank you very much for your valuable suggestion. We originally believed that the research focus of this paper was on predicting the remaining life of the blades while adjusting the controller parameters based on the prediction results. This would effectively reduce the fatigue load on the relevant components and increase their service life. The focus of the research is on the study of control strategies, as the operation and maintenance strategy of wind turbines is a complex problem that involves cost optimization of the entire wind farm. We had planned to conduct a more detailed study in our next paper. However, after reading your review comments, we believe that adding some calculations regarding maintenance costs and the number of unit starts and stops in this article is more reasonable. Therefore, we have calculated the maintenance costs, specifically as follows:
In addition to the above simulations, in order to verify the practical value of the proposed strategy, some research on unit operation and maintenance cost calculation should be considered in the paper. We compared two commonly used operation and maintenance strategies and the operation and maintenance costs of the strategy proposed in this article. The simulation results showed that using the strategy proposed in this article for control can effectively reduce the operation and maintenance costs of the unit. The total cost generated during the operation of wind turbines can usually be described using equation (44): (44) In equation(44), representing the total cost, representing the inspecting cost, representing the preventive intervention cost, representing the corrective intervention cost and representing the cost of electricity production losses, which depends on downtime and input wind speed.
Assuming the initial electricity price is a constant binit, the analysis of the cycle life model ignores inflation and electricity price fluctuations in practice. The impact and benefits generated belong to the expected electricity revenue B, which is used here to calculate the production loss Closs caused by downtime. (45) In equation (45), binit representing the electricity price constant, Th,stop is the current inspection and repair related downtime, P(v) is the output power of wind turbine.
Maintenance measures are divided into state-based preventive repair and fault-based corrective repair. If the crack length detected during the inspection is greater than the repair reference value arep, state-based preventive maintenance is performed on the same day as the test, and the repair reference value arep should be much lower than the fault threshold afall, so that a major failure will not occur due to a larger crack before the next inspection, that is, preventive repair is performed when the ains >arep. Repair measures A: only take corrective repair, which requires more than 12 maintenance personnel and 240h downtime, and the replacement cost of the blade is 632000 yuan; repair measure B: take corrective repair at the same time, and combined with preventive repair based on inspection conditions, preventive maintenance requires more than 3 maintenance personnel and 60 hours of downtime, material cost required for preventive repair of leaves is one tenth of the replacement cost. The preventive intervention cost Crep and corrective intervention cost Ccor generated by each maintenance strategy need to be combined with the transportation strategy and the downtime Th, stop is used for calculation. According to the guidance price for offshore wind power approved by the National Development and Reform Commission of China in 2019, it has been adjusted to 0.75 yuan/kW · h per kilowatt hour. Based on the downtime, the cost of electricity production loss, Closs, can be calculated. Assuming that weather conditions with wind speeds less than 10m/s can be found, and inspection and maintenance will not be interrupted by adverse weather conditions, the downtime Th, stop can be expressed as the time required for inspection or maintenance. Calculate the single inspection cost Cins, preventive repair cost Crep, and corrective cost Ccoren based on relevant information. These costs include wind energy damage cost Closs, transportation cost Ctran, and maintenance personnel's individual cost Cpeo. The transportation cost Ctran is calculated based on the inspection and repair time performed each time. The number of days required for the transport ship Td and stop can be calculated based on the number of downtime Th and stop. The effective working hours for maintenance work are 8 hours per day. The cost calculations for each item are shown in Table 2. Table 2.The various O&M costs and calculation methods of single wind turbine
The maintenance methods for wind turbines usually include maintenance based on corrective repair and maintenance based on two types. The maintenance strategy based on corrective repair does not actively intervene in the crack propagation of the blades, and only initiates corrective repair when a> afall causes major faults. Due to the lack of inspection and preventive repair measures, the total expected maintenance cost Ctotal only needs to calculate the transportation cost required for corrective repair, maintenance cost of operation and maintenance personnel, downtime cost, and blade replacement cost. According to our calculations, the total cost expenditure based on corrective maintenance and time related expenses is approximately 5.94 million yuan. The maintenance strategy based on preventive repair is usually based on the results of regular inspections of high-risk failure nodes, combined with the current control strategy of the unit for preventive maintenance. The calculation of expected maintenance costs only includes transportation costs, inspection costs for operation and maintenance personnel, downtime costs, and preventive maintenance costs for blades. Therefore, the expected maintenance cost Ctotal for wind turbine blades is approximately 4.94 million yuan. The following figure describes the relationship between crack length and time of three blades based on preventive repair strategy:
From the figure, it can be seen that when the cracks on the three blades reach A, B, and C, maintenance is required. At this time, the maintenance cost is calculated, totaling 4.94 million yuan. The PF based prediction proposed in this article can effectively predict the trend of blade crack propagation, accurately detect cracks during inspection, and combine with blade life control strategies to unify the crack propagation rate of each blade at designated inspection time points, facilitating centralized maintenance and reducing maintenance costs. The crack propagation at node 1 of blade 1 under the preventive maintenance strategy based on the integrated control strategy of blade life is shown in the following figure, and the expected minimum total cost throughout the entire life cycle is about 1.59 million. It can be seen that combining PF prediction can help maintenance personnel effectively identify the trend of blade crack propagation in the presence of historical observation values, and carry out maintenance work in a timely manner when the blades reach the repair reference value arep. At the same time, combined with the life integration control strategy, it can effectively ensure that the length of each blade crack during the preventive maintenance cycle does not exceed the repair reference value arep as much as possible, avoiding maintenance personnel from carrying out preventive repairs on each blade in stages, reducing the total repair time required for all blades of the unit throughout the entire life cycle, and lowering the expected total cost. The expected costs and total costs of corrective maintenance strategy, preventive maintenance strategy, and preventive maintenance strategy based on life integrated control are compared in the following table. Among them, Closs is an uncertain cost, which is more significant in corrective maintenance and basic preventive maintenance with longer downtime due to the influence of wind speed and downtime. As for unplanned downtime, it is related to the issue of sudden blade failures. We are still researching related content, such as how to optimize the operation and maintenance costs of the entire wind farm and reduce the number of unit starts and stops. We will conduct more in-depth research on these issues in future studies.
We sincerely appreciate your feedback. However, regarding the research on unit operation and maintenance strategies, we are currently conducting it and will provide more detailed explanations in future work.
Comments 4: Suggestions for future research could include exploring the scalability of the control strategy to different types of wind turbines or environmental conditions. Additionally, investigating the integration of machine learning techniques could further enhance predictive capabilities. Response 4: We sincerely thank you for pointing out the direction for our future research. After carefully reading your review comments, we have also been inspired and will write your research content as our future research direction after the conclusion of the paper. Thank you again for your valuable suggestions, which have really helped improve the quality of our paper.
Our future research may include exploring the scalability of control strategies for different types of wind turbines or environmental conditions, as well as qualitative research on how to further improve life prediction capabilities. For example, integrating machine learning techniques or processing redundant data before predicting the lifespan of wind turbines can improve prediction performance to a certain extent.
|

Round 2
Reviewer 2 Report
Comments and Suggestions for Authors
It can be accepted now.
Author Response
Comments 1:It can be accepted now.
Response 1: We greatly appreciate your recognition of this work.
Reviewer 4 Report
Comments and Suggestions for Authors
The revision looks fine. However, the main contributions should be improved.
Author Response
Comments 1:The revision looks fine. However, the main contributions should be improved.
Response 1:
Response 1: We sincerely appreciate your review comments and have made revisions to the paper according to your suggestions. The specific modifications are as follows:
Based on the above analysis, we can find that the current methods for life prediction both domestically and internationally are mainly based on data-driven research. However, there are few methods that combine blade life with unit control. In fact, while conducting life prediction, adjusting the controller parameters of the unit based on the life prediction results can effectively increase the remaining life of relevant important subsystems.
The inner loop load shedding controller is mainly designed based on random disturbance correction control to balance the load shedding capability of the controller and the workload of the pitch actuator, while meeting the predefined blade life of the operation and maintenance strategy, avoiding unplanned shutdowns, and reducing the total maintenance cost of the unit.
We also highlighted the contribution of our work in the penultimate paragraph of the paper, specifically: |
The main contributions of this article include: 1.We established a crack propagation model for blades based on the Pair's law and applied the rain flow counting method to solve the cyclic stress on the blades in order to obtain the strength factor in the model. 2. We propose a Bayesian based method for predicting the remaining life of blades, which addresses the issue of infinite integration in Bayesian filters. We implement it using particle filtering algorithm, which provides a suboptimal solution for the Bayesian estimator through Monte Carlo integration. Simulation shows that although particle filtering has a high time cost, it has more advantages in terms of lifetime distribution and prediction stability. 3. We designed an SDAC controller and adjusted the controller parameters based on the predicted lifespan, effectively reducing the fatigue of the pitch control actuator and extending its service life. In response to the presence of untreated process noise and measurement noise in traditional interference correction control, we have adopted the Kalman filter method to improve the robustness of the controller. 4. We calculated the operation and maintenance costs under traditional control strategies and our proposed strategy, and found that adopting the strategy proposed in this paper can effectively reduce the operation and maintenance strategies of the unit, which has practical engineering significance.
|